# Three types of genes underlying the *Gametophyte factor1* locus cause unilateral cross incompatibility in maize

Yuebin Wang ®[1,6], Wenqiang Li ®[1,6], Luxi Wang[1], Jiali Yan[1], Gang Lu ®[1], Ning Yang[1,2], Jieting Xu[1], Yuqing Wang ®[1], Songtao Gui[1], Gengshen Chen ®[1], Shuyan Li ®[1], Chengxiu Wu ®[1], Tingting Guo[1,2], Yingjie Xiao[1,2], Marilyn L. Warburton ®[3], Alisdair R. Fernie ®[4], Thomas Dresselhaus[5] & Jianbing Yan ®[1,2] ✉

Unilateral cross incompatibility (UCI) occurs between popcorn and dent corn, and represents a critical step towards speciation. It has been reported that *ZmGa1P*, encoding a pectin methylesterase (*PME*), is a male determinant of the *Ga1* locus. However, the female determinant and the genetic relationship between male and female determinants at this locus are unclear. Here, we report three different types, a total of seven linked genes underlying the *Ga1* locus, which control UCI phenotype by independently affecting pollen tube growth in both antagonistic and synergistic manners. These include five pollen-expressed *PME* genes (*ZmGa1Ps-m*), a silk-expressed PME gene (*ZmPME3*), and another silk-expressed gene (*ZmPRP3*), encoding a pathogenesis-related (PR) proteins. *ZmGa1Ps-m* confer pollen compatibility. Presence of *ZmPME3* causes silk to reject incompatible pollen. *ZmPRP3* promotes incompatibility pollen tube growth and thereby breaks the blocking effect of *ZmPME3*. In addition, evolutionary genomics analyses suggest that the divergence of the *Ga1* locus existed before maize domestication and continued during breeding improvement. The knowledge gained here deepen our understanding of the complex regulation of cross incompatibility.

Reproductive isolation is one of the major driving forces of speciation and evolution[1,2], and is generally classified as either prezygotic or postzygotic[3]. The nature of reproductive isolation in angiosperms involves multiple aspects, including pollen recognition, pollen tube growth, pollen tube reception and the fusion between male and female gametes[4–6]. Unilateral cross incompatibility (UCI) is a type of prezygotic reproductive isolation that occurs in maize between popcorn and dent corn varieties. Since it first description in 1902[7], a linked locus named *Gametophyte factor1* (*Ga1*) has been reported on the short arm

of chromosome 4[8,9]. It was further reported that the *Ga1* locus is composed of three different haplotypes, *Ga1-S*, *Ga1-M*, and *ga1*. Popcorn possesses the *Ga1-S* haplotype, and dent corn possesses the *ga1* haplotype. *ga1* plants accept any type of maize pollen; *Ga1-S* plants cannot be pollinated by *ga1* pollen, but can pollinate all other genotypes; while *Ga1-M* plants exhibit wide compatibility in either corn type[10].

Previous studies have demonstrated that UCI is caused by failed growth of *ga1* pollen tubes. After successful germination, *ga1* pollen

[1]National Key Laboratory of Crop Genetic Improvement, Huazhong Agricultural University, Wuhan 430070, China. [2]Hubei Hongshan Laboratory, Wuhan 430070, China. [3]USDA ARS Corn Host Plant Resistance Research Unit, Mississippi State, MS 39762, USA. [4]Department of Molecular Physiology, Max-Planck-Institute of Molecular Plant Physiology, 14476 Potsdam-Golm, Germany. [5]Cell Biology and Plant Biochemistry, University of Regensburg, 93053 Regensburg, Germany. [6]These authors contributed equally: Yuebin Wang, Wenqiang Li. ✉e-mail: yjianbing@mail.hzau.edu.cn

tubes arrest growth before reaching the ovules in *Ga1-S* silks[11]. Efforts to genetically dissect the *Ga1* locus revealed that UCI is governed by at least two determinants: (i) a male determinant acting in pollen/pollen tubes and endows pollen tubes to become compatible for any type of silks. (ii) a female determinant acting in silk, enabling silk to reject pollen tubes that lack male determinant (incompatibility)[12,13]. Recently, *ZmGa1P*, encoding a pectin methylesterase, which is highly expressed in *Ga1-S* and *Ga1-M* pollen, has been shown as a male determinant of the *Ga1* locus[14]. Moreover, *ZmPME3*, was identified and shown to be highly expressed in *Ga1-S* but barely in *ga1* silks[15]. It shares high similarity with *Tcb1-f*, a gene identified in teosinte and proved to be responsible for blocking maize pollen tubes[16]. Therefore, *ZmPME3* was suggested as a likely candidate for the female determinant. However, how the female determinant functions, and the genetic relationship between male and female determinants at the *Ga1* locus[17-19] still remained elusive.

In this study, we use genetic analysis to separate the *Ga1* locus into two functional components and identify seven causal genes, encoding three types of proteins. Transgenic approaches confirm that the seven genes harbor both male and female determinant functions, and work together to constitute the UCI system. Moreover, the three types of genes exhibit clear variation between cultivated, landrace and wild maize and thus provides an excellent system to deeply understand the history of maize domestication and improvement.

## Results

### The *Ga1* locus contains two functional components

To confirm that UCI in our genetic material is caused by abnormal pollen tube growth[8,15], rather than failed gamete fusion (Fig. 1a), we collected SK (*Ga1-S*) silks pollinated by Zheng58 (Z58, *ga1*) pollen and self-pollinated SK silks at 8 h and 24 h after pollination, respectively. We observed that Z58 pollen tubes were significantly shorter than those of SK pollen tubes in SK silks [$2.07 ± 0.27$ cm vs. $3.80 ± 0.34$ cm ($P = 3.91E\text{-}15$) at 8 h; $3.55 ± 0.72$ cm vs. $12.49 ± 0.86$ cm ($P = 9.90E\text{-}25$) at 24 h)] (Supplementary Fig. 1a, b). All observed SK pollen tubes were equivalent in length to the SK silks, By contrast, the longest pollen tubes of Z58 reached only about 30% of the total length of SK silks at 24 h after pollination, with a heavy callose deposition at the tip and arrest growth before reaching the ovule[11] [$28 ± 5\%$ vs. 100% ($P = 4.35E\text{-}30$)] (Supplementary Fig. 1c, d). When SK silks were cut shorter than 5 cm, we were able to harvest a few kernels pollinated by Z58 (Supplementary Fig. 1e), implying that UCI is caused by abnormal Z58 pollen tube elongation rather than failed gamete fusion. This observation strongly suggests that promoting pollen tube growth is the key to overcome incompatibility.

Next, a Recombination Inbred Line (RIL) population between Z58 × SK was developed, and genotyped using high-density markers[20,21]. We detected a segregation distortion locus that the genotypes of two parents are significantly deviating from the expected ratio (1:1) on chromosome 4 from 2 Mb to 25 Mb (Supplementary Fig. 2), which overlapped with the defined the *Ga1* locus in previous reports[8,22-24]. Approximately 4900 individuals from the Z58 × SK $F_2$ population were subjected to fine map the *Ga1* locus using genotypic segregation distortion as a phenotype. Fine mapping results revealed that the *Ga1* locus can be separated into two components that both influenced segregation distortion, we named here as Component1 (between marker M4 and M7) and Component2 (between marker M7 and M9) (Fig. 1b and Supplementary Fig. 3). When individuals possessed the SK genotype at Component1, segregation distortion completely disappeared (R5, $P = 0.46$, $n = 197$; Fig. 1b), suggesting that Component1 is a major locus for overcoming cross incompatibility. Component2 affected the degree of segregation distortion only if Component1 was heterozygous. When Component2 was homozygous for the SK genotype, Component1 showed the strongest segregation distortion (R2, $P = 2.66E\text{-}17$, $n = 164$, Fig. 1b). When Component2 was

heterozygous, Component1 showed lighter segregation distortion (R1, $P = 7.25E\text{-}06$, $n = 230$; R4, $P = 6.90\text{-}08$, $n = 289$, Fig. 1b). When Component2 was homozygous for the Z58 genotype, Component1 did not show segregation distortion (R3, $P = 0.64$, $n = 199$, Fig. 1b). To further confirm function of the two components, we used a diverse Association Mapping Panel (AMP)[25] as male parents to pollinate CML304 (*Ga1-S*), and took the CML304 seed set ratio as a phenotype for a genome-wide association study (GWAS)[26] (see Materials and Methods for details). SNPs significantly associated with seed set ratio were identified within the Component1 region (Supplementary Fig. 4). These results demonstrated that Component1 contains a male determinant that enables pollen tubes to overcome the block effect of *Ga1-S* silks, and that the function of Component2 depends on Component1. Component2 is most likely being a weak regulator affecting UCI.

### Genomic structure analysis and identification of candidate genes at the *Ga1* locus

Traditional fine mapping to clone the *Ga1* locus has been pursued for years, the interval cannot be further narrowed below 1.7 Mb, suggesting that huge structural variations between the *Ga1-S* and *ga1* genomes may not allow genetic mapping. A high-quality SK genome has recently become publicly available[27]. Although the Z58 genome is still not available, B73 (*ga1*) shows the same UCI phenotype as Z58. We were thus able to compare the genomic sequences of the *Ga1* locus between B73[28] and SK. Despite a similar size of the *Ga1* locus in the B73 and SK genomes, a very different sequence, with few syntenic blocks between markers M4 and M7 was detected (Fig. 1c and Supplementary Fig. 4e). This explains why it was so difficult to find sufficient recombinant individuals for fine mapping strategies.

In the region of Component1, 28 genes were annotated within the SK genome. These include 13 consecutive *PECTIN METHYLESTERASE* (*PME*) genes (Supplementary Data 1). However, in the corresponding Component1 region of B73, there is only one annotated gene, *Zm00001d048936*, (also known as the *ga1* type *ZmGa1P*), highly expressed in pollen and exhibiting synteny to the *PME* gene cluster (Fig. 1c, d and Supplementary Data 2). RNA-seq data from nine SK tissues further confirmed that five of the thirteen *PME* genes are highly expressed in SK pollen (*ZmGa1Ps-m*: *ZmGa1P.1*, *ZmGa1P.2*, *ZmGa1P.3*, *ZmGa1P.4*, and *ZmGa1P.5*), and one gene (*ZmPME3*) is highly expressed in SK silks (Supplementary Fig. 5).

A pollen-expressed *PME* gene at the *Ga1* locus, *ZmGa1P*, was found in SDGa25 (*Ga1-S*) and previously reported to be responsible for cross incompatibility by controlling the equilibrium of pectin esterification / de-esterification at the apical region of the growing pollen tube[14,29]. We found that *ZmGa1Ps-m* genes not only exhibit high sequence identity to one another (identity > 95%), but also to *ZmGa1P* (identity > 95%), with only a few single nucleotide polymorphisms (SNPs) (Supplementary Fig. 6). Most of these cause synonymous mutations (Supplementary Fig. 7). When comparing *ZmGa1Ps-m* genes with *Zm00001d048936*, which has been proved as a non-functional homologous gene of *ZmGa1P* in the B73 genome, we found that *Zm00001d048936* (*ga1*) harbors a nonsense point mutation from G to A (Supplementary Fig. 8), producing premature termination (Supplementary Fig. 9), consisting with previous findings[14].

To further ascertain if *ZmGa1Ps-m* genes contribute to regulating pollen tube growth, we extracted RNA from pollen of SK (*Ga1-S*), P178 (*Ga1-M*) and Z58 (*ga1*) and determined the expression level of the *PME* gene cluster (Supplementary Data 3). SNPs in the RNA-seq data clearly distinguished *ZmGa1Ps-m* genes from each other (Supplementary Figs. 10 and 11) and revealed their respective high expression in SK and P178 pollen, but were absent in Z58 pollen (Fig. 1e and Supplementary Fig. 12). This finding indicates that all *ZmGa1Ps-m* genes are unique genes, likely required to overcome the *Ga1-S* barriers. Moreover, sequence alignment between SK and B73 revealed that ~800 bp coding sequence of the each *ZmGa1Ps-m* genes was missing across the entire

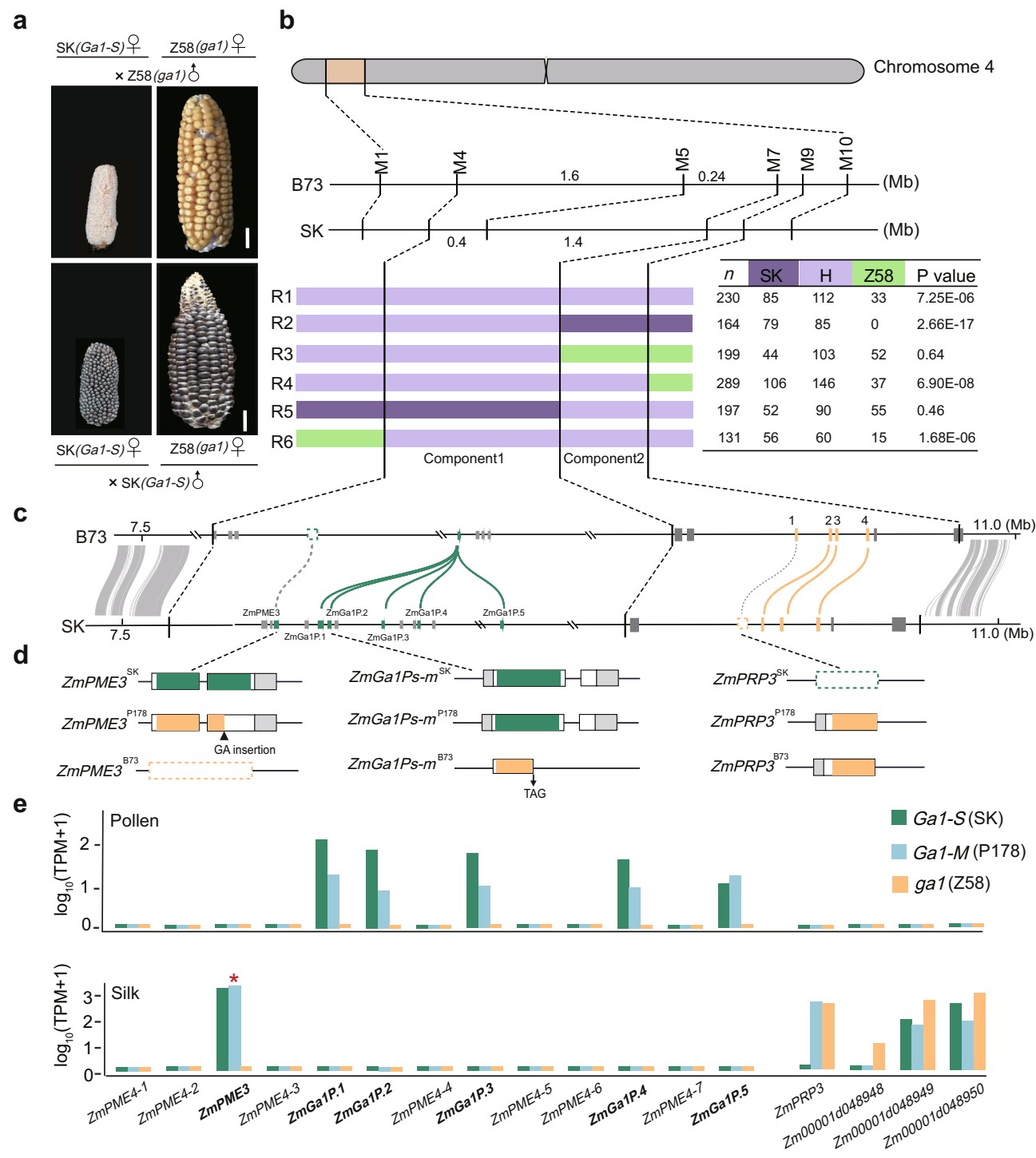

B73 genome. A similar result was also observed in the HP301(*Ga1-S*) genome[30] (Supplementary Fig. 13), indicating a presence-absence variation (PAV)[31]. Based on the CML304 (*Ga1-S*) seed set ratios crossed with AMP pollen, and the seed set ratios from crosses by using AMP lines each as a female parent and Z58 (*ga1*) as a male parent, AMP inbred lines could be divided into three haplotypes (see Materials and Methods for details): *ga1* (*n* = 303), *Ga1-M* (*n* = 82), and *Ga1-S* (*n* = 7). After counting mapped reads from AMP whole genome re-sequencing data[27], we found that *ZmGa1Ps-m* genes were absent in all *ga1* but present in *Ga1-M* and *Ga1-S* lines. This demonstrated that PAV of *ZmGa1Ps-m* genes is widespread at the population level (Supplementary Fig. 14). Therefore, we assumed that there are also multiple copies of *ZmGa1P* in the SDG25a genome. However, limited by the length of

the BAC clone, it is hard to assemble all the peptide fragment from multiple highly similar *ZmGa1P* genes in an interval close to 1.5 Mb. Silk-expressed *ZmPME3* has been reported as a candidate female gene involved in UCI[15]. We also observed that *ZmPME3* is the only *PME* gene within the Component1 region and that it is highly expressed in SK silks (Fig. 1e and Supplementary Fig. 5), but absent from the entire B73 genome (Fig. 1c, d). However, previous studies also pointed out that *ZmPME3* has a frame-shift mutation in some *Ga1-M* lines, while a few lines possess intact *ZmPME3* transcripts[15,32], which makes it difficult to conclude whether *ZmPME3* harbors female determinant function. In our next investigation, we extracted RNA from silk of SK (*Ga1-S*), P178 (*Ga1-M*), Z58 (*ga1*) and 70 selected inbred lines from the AMP, containing *ga1* (*n* = 34), *Ga1-M* (*n* = 30) and *Ga1-S* (*n* = 6) haplotypes. Both

**Fig. 1 | Fine mapping and genomic structure of the *Ga1* locus in maize. a** The UCI phenotype: self-pollinated SK (*Ga1-S*), Zheng58 (Z58, *ga1*) and reciprocal crosses between SK and Z58. Scale bar = 2 cm. **b** *Ga1* was fine mapped to a 2 Mb interval, flanked by markers M4 and M9. It contains two functional components, Component1 (M4-M7) and Component2 (M7-M9). SK represents the SK genotype, Z58 represents Zheng58 genotype and H represents heterozygous genotype. The numbers in each row represent the number of individuals in the progeny generated after self-fertilization of recombinant plants with the genotype on the left. *P* value represents the significance of the chi-square test. **c** Sequence alignment of the *Ga1* locus between B73 and SK genomes. Syntenic blocks are highlighted by gray lines. Five *ZmGa1Ps-m* genes, one silk-expressed gene *ZmPME3*, and four silk-expressed *CRP* genes are highlighted by green and orange boxes. Alignments of homologous genes on the B73 and SK genome are indicated by green and orange lines. Present-absent variations are indicated by gray dotted lines and absent genes are indicated by dotted boxes. **d** Gene structure alignment of *ZmGa1Ps-m* genes, silk-expressed gene *ZmPME3* and *ZmPRP3* compared between SK (*Ga1-S*), P178 (*Ga1-M*) and B73 (*ga1*). The complete pectin methylesterase domains in *ZmPME3^SK* genes are highlighted in green. The incomplete pectin methylesterase domains in *ZmPME3^P178* that is interrupted by a 2 bp insertion is highlighted in orange. Absent *ZmPME3^B73* is indicated by orange dotted boxes. The complete pectin methylesterase domains in *ZmGa1Ps-m^SK* and *ZmGa1Ps-m^P178* are highlighted in green. The incomplete pectin methylesterase domain in *ZmGa1Ps-m^B73* that due to a nonsense point mutation from G to A is highlighted in orange. The barwin domains in *ZmPRP3^P178* and *ZmPRP3^B73* are highlighted in orange boxes. Absent *ZmPRP3^SK* is indicated by green dotted boxes. White boxes represent coding sequence. UTR region is highlighted with gray boxes. **e** Expression patterns of thirteen *PME* genes, *ZmPRP3*, *Zm00001d048948*, *Zm00001d048949*, and *Zm00001d048950* in silk and pollen of SK (*Ga1-S*), P178 (*Ga1-M*) and Z58 (*ga1*) as determined by RNA-seq data. TPM, Transcripts Per kilobase per Million mapped reads. "*" highlight with red presents non-functional transcript of *ZmPME3* in *Ga1-M* silk.

RNA-seq (Supplementary Data 4) and quantitative RT-PCR analysis (Supplementary Data 5) confirmed that *ZmPME3* is expressed in silks of all selected *Ga1-S* and a proportion of *Ga1-M* lines, but conversely is not expressed in any silks of *ga1* lines (Fig. 1e and Supplementary Fig. 15a). Sequencing of *ZmPME3* transcripts from selected *Ga1-M* and *Ga1-S* silks implied that the coding sequences derived from *Ga1-M* silks is interrupted by a 1 or 2 bp insertion (Supplementary Fig. 15b), which is consistent with the previous study[15]. It thus appears that *ZmPME3* generates functional transcripts with abundant expression in all *Ga1-S* lines, which makes *ZmPME3* at least consider as a candidate female determinant of the *Ga1-S* type, enabling *Ga1-S* silks to block *ga1* pollen tube growth. We next analyzed expression levels of the annotated genes in the Component2 region using pollen and silks of SK (*Ga1-S*), P178 (*Ga1-M*) and Z58 (*ga1*). Notably, this region contains four genes, encoding cystein-rich proteins (*CRPs*)[33,34]. One of them, *Zm00001d048947*, which encodes a small protein containing a barwin domain (known as *Zea mays pathogeneesis-related protein3 ZmPRP3*[35]), is absent from the entire SK genome (Fig. 1d), but is highly expressed in Z58 and P178 silks (Fig. 1e and Supplementary Fig. 16). Moreover, 20 × re-sequencing data of inbred lines from AMP further confirmed that *ZmPRP3* is absent from all *Ga1-S* lines, but present in 90% of *Ga1-M* lines and 99% of *ga1* lines (Supplementary Fig. 17). Quantitative RT-PCR results (Supplementary Fig. 15a) further showed that *ZmPRP3* is expressed in all tested *ga1* and *Ga1-M* lines, and contains a highly conserved peptide sequence with five cysteines (Supplementary Fig. 18 and Supplementary Table 1). Considering the key roles of *CRPs* in plant defense and reproduction[36–38], we thus speculated that *ZmPRP3* may serve as a positive regulator to accelerate pollen tube growth in *ga1* and *Ga1-M* silks, The other three *CRP* genes (*Zm00001d048948*, *Zm00001d048949*, and *Zm00001d048950*) display high sequence similarity (identity > 80%) to *ZmPRP3* (Supplementary Fig. 19), but exhibit different expression pattern (Fig. 1e). While *Zm00001d048948* is only expressed in *ga1* silks, *Zm00001d048949* and *Zm00001d048950* are expressed in all genotypes, we thus considered only *ZmPRP3* to be the best candidate of a causal gene of Component2.

**Transgenic experiments support the causal roles of identified *ZmGa1Ps-m*, *ZmPME3*, and *ZmPRP3* in regulating pollen tube growth**

To verify whether *ZmGa1Ps-m* genes all regulate pollen tube growth, we over expressed each *ZmGa1Ps-m* gene individually using the ubiquitin promoter in the KN5585 (*ga1*) background[39]. However, *PME* gene expression level in the transgenic over-expression lines reached 2% to 13% of that found in SK pollen (Fig. 2a), which is lower than that of *ZmGa1P* in B104 (*ga1*) pollen (a 6287 bp genomic fragment containing *ZmGa1P* and its native promoter was cloned from SDGa25 and introduced into the inbred line B104 to verify *ZmGa1P* function)[14]. We speculate this discrepancy is caused because the ubiquitin promoter is weaker in pollen compared with the *ZmGa1P* promoter. At 8 h after pollination, the maximal length of pollen tubes that over-expressed *ZmGa1Ps-m* genes were significantly longer than those of the wild type [WT, KN5585 (*ga1*)] in SK silks, ranging from 1.46 times longer for *ZmGa1P.4* [1.69 ± 0.19 cm vs. 1.15 ± 0.29 cm (*P* = 2.40E-08)] to 1.81 times longer for *ZmGa1P.5* [2.08 ± 0.31 cm vs. 1.15 ± 0.29 cm (*P* < 2.00E-16)]. However, they only reached 44.5–54.7% of the length of SK pollen tubes [*ZmGa1P.4*, 1.69 ± 0.19 cm vs. 3.92 ± 0.41 cm (*P* < 2.00E-16); *ZmGa1P.5*, 2.08 ± 0.31 cm vs. 3.92 ± 0.41 cm (*P* < 2.00E-16)] (Fig. 2b). In summary, there is a clear positive correlation between pollen tube length and *ZmGa1Ps-m* genes expression levels. As a consequence, SK plants fertilized with pollen from transgenic lines over-expressing *ZmGa1P.2* exhibited a significant increase in seed number compared with WT pollen [12.6 ± 7% vs. 0.00 (*P* = 0.0043)] (Fig. 2c, and Supplementary Fig. 20). These data together indicate that *ZmGa1Ps-m* are causal genes, and that their accumulative expression level accelerate *Ga1-S* and *Ga1-M* pollen tube growth, which ultimately lead pollen tubes to overcome the *Ga1-S* barrier.

We also over-expressed *ZmPME3* transcript derived from the SK genome in the KN5585 background to investigate whether *ZmPME3* is capable of blocking *ga1* pollen tube growth. Although transgenic plants maintained a lower *ZmPME3* expression levels in silks than SK (reaching only about 2 ~ 10% of the SK level), they expressed far more *ZmPME3* transcripts than WT [KN5585 (*ga1*)]. This led to a slower KN5585 (*ga1*) pollen tube growth rate in silks of three over-expressed *ZmPME3* plants, at 8 h after pollination [2.17 ± 0.38 cm, 2.04 ± 0.42 cm and 2.14 ± 0.42 cm vs. 4.17 ± 0.23 cm (*P* < 2.00E-16)]. However, the longest KN5585 (*ga1*) pollen tubes in silks of over-expressed *ZmPME3* plants were still nearly two times longer than those in SK silks [2.17 ± 0.38 cm, 2.04 ± 0.42 cm and 2.14 ± 0.42 cm vs. 1.15 ± 0.29 cm (*P* = 8.30E-13; *P* = 8.30E-12; *P* = 8.30E-15)] (Fig. 2d, e). This suggests that *ZmPME3* is a causal female determinant gene of UCI in the Component1 region, conferring silks to reject *ga1* pollen. Its expression level is negatively related to the growth rate of *ga1* pollen tubes. Notably, we observed that KN5585 (*ga1*) pollen crossed onto *ZmPME3^OE+* lines resulted in full seed set (Fig. 2f and Supplementary Fig. 20). This may be due to insufficient expression of *ZmPME3* in silks, or there are other factors that exist in *ga1* silks and rescue the cross incompatibility phenotype.

To further investigate the roles of *ZmPRP3* identified in the Component2 region in regulating pollen tube growth, *ZmPRP3* and its three homologs were each knocked out by CRISPR/Cas9 in KN5585 inbred lines to yield the corresponding homozygous null mutants, referred to here as *CR-Zmprp3*, *CR-Zm00001d048948*, *CR-Zm00001d048949* and *CR-Zm00001d048950*. Genome sequencing identified a 147 bp deletion in the first exon of *ZmPRP3* (Fig. 2g), resulting in a deletion of 49 amino acids that included the barwin domain. At 8 h after pollination, KN5585 (*ga1*) pollen tubes grew significantly slower in the *CR-Zmprp3* silks compared to WT [KN5585 (*ga1*)] silks [2.16 ± 0.29 cm vs. 4.18 ± 0.23 cm (*P* = 1.51E-26)] (Fig. 2h).

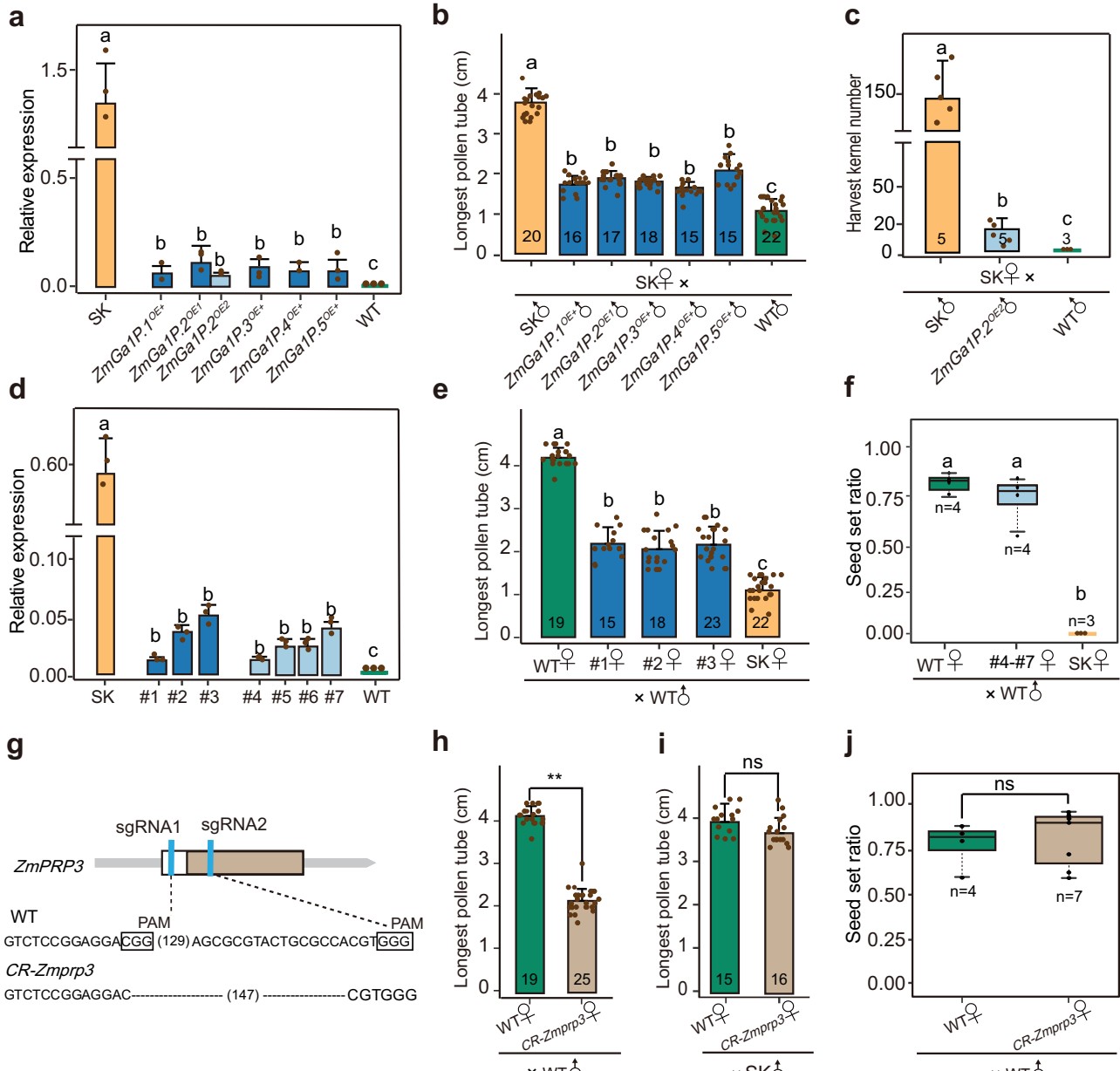

**Fig. 2 | Transgenic confirmation that *ZmGa1Ps-m*, Zm*PME3* and *ZmPRP3* regulate pollen tube growth. a** Quantitative expression analysis of *ZmGa1Ps-m* genes in pollen of over-expressed transgenic plants, WT [KN5585 (*ga1*)] and SK. **b** Comparison of the longest pollen tubes in SK silks at 8 h after pollination, using pollen of WT, SK, and over-expressed lines for each *ZmGa1Ps-m* genes. **c** Statistical analysis of SK seed set ratio crossed with SK, WT, *ZmGa1P.2* over expressed lines pollen. **d** Quantitative expression analysis of *ZmPME3* in silks of *ZmPME3* over expressed lines, WT and SK. #1-#3 represents three *ZmPME3* over expressed transgenic plants. #4-#7 represents four *ZmPME3* over expressed transgenic plants. **e** Comparison of the longest pollen tubes of WT [KN5585 (*ga1*)] at 8 h after pollination in silks of *ZmPME3* over expressed, WT and SK plants. **f** Statistical analysis of the seed set ratio from crosses by using SK, WT, and *ZmPME3* over expressed plants each as a female parent and WT [KN5585 (*ga1*)] as a common male parent. **g** Sequence of *ZmPRP3* in WT and CRISPR-Cas9 edited plants. Lines in blue denote the sgRNA, and the protospacer-adjacent motif (PAM, NGG) is indicated by the black square. White box indicates the coding sequence and barwin is highlighted with brown boxes. **h** Comparison of the longest pollen tubes of WT [KN5585 (*ga1*)] at 8 h after pollination in silks of WT and *CR-Zmprp3* plants. **i** Comparison of the longest pollen tubes of SK at 8 h after pollination in silks of WT and *CR-Zmprp3* plants. **j** Statistical analysis of seed set ratio from crosses by using *CR-Zmprp3* and WT each as a female parent and WT [KN5585 (*ga1*)] as a common male parent. For **a**–**f**, data were analyzed according to the LSD test. For **h**–**j**, data were analyzed by two-tailed Student's *t*-test. Error bars represent mean + SD (**a**–**e**, **h**, **i**). a, b, c indicate significant differences ($P < 0.01$) (**a**–**f**). Asterisks indicate significant differences (**$P < 0.01$) (**h**), ns (not significant, $P > 0.05$) (**j**). Relative expression ($n = 3$, biologically independent samples) (**a**, **d**). Longest pollen tube length or harvest kernel number ($n$ = number of biologically independent samples in each column) (**b**, **c**, **e**). In box plots, the center line represents the median, box edges delimit lower and upper quartiles and whiskers show the highest and lowest data points, n present sample size (**f**, **j**). Source data are provided as a Source Data file.

With the exception of *CR-Zmprp3*, none of the *Zm00001d048948*, *Zm00001d048949* and *Zm00001d048950* knock-out lines affected the KN5585 pollen tube growth rate (Supplementary Figs. 21 and 22). Furthermore, SK pollen tubes showed comparable growth rates in both *CR-Zmprp3* and WT silks [$3.71 \pm 0.35$ vs. $3.99 \pm 0.33$ ($P = 0.034$)] (Fig. 2i). We thus conclude that only *ZmPRP3* contributes to regulating *ga1* pollen tube growth, but it does not influence *Ga1-S* pollen tubes. KN5585 pollen crossed onto *CR-Zmprp3* resulted in successful crosses (Fig. 2j and Supplementary Fig. 20), demonstrating that *ZmPRP3* alone is not sufficient to regulate UCI.

RNA-Seq analysis on silks of Recombination Lines (RL) and *ZmPRP3* knock-out lines were next used to elucidate which pathway might be controlled by *ZmPRP3*. 305 differentially expressed genes (DEGs) with fold change >2 were detected in lines expressing *ZmPRP3* compared to those lacking *ZmPRP3*. The DEGs were associated with 300 different Gene Ontology (GO) terms and the most enriched term was related to metal ion binding ($n = 55$) (Supplementary Fig. 23). RNA-seq analysis in RL silks expressing *ZmPME3* compared to those lacking *ZmPME3* identified 10 DEGs, from which *ZmPME3* itself exhibited the highest fold change (Supplementary Fig. 24). After comparing the two different groups of DEGs, we found that the absence of *ZmPRP3* did not cause change in expression level of any *PME* genes. The degree of methylesterification (DM) of silk cell walls in *ZmPME3* over-expression lines (*ZmPME3^OE+^*), *ZmPRP3* knock-out lines (*CR-Zmprp3*), SK and WT [KN5585 (*ga1*)] were also compared using LM20 and LM19, antibodies that preferentially recognizes methylesterified pectin[40,41] and low-methylesterified homogalacturonans (HGs)[42] respectively. SK and *ZmPME3^OE+^* silk cells exhibited weaker fluorescence intensity compared to the WT and *CR-Zmprp3* using LM20 antibody (Supplementary Fig. 25). On the contrary, the fluorescence intensity of cells appeared stronger in *CR-Zmprp3* and WT silks using LM19 antibody (Supplementary Fig. 26). These results indicated that the DM of cells in SK and *ZmPME3^OE+^* are lower than that of WT and *CR-Zmprp3*. The fluorescence intensity of *CR-Zmprp3* and WT silk cells have no difference under two conditions, suggesting *ZmPRP3* does not affect the DM of silk cells. Altogether, this indicates that there exist additional mechanisms regulated by *ZmPRP3* and differ from *ZmPME3* activity, which also contribute to the regulation of UCI.

## Pollen and silk expressed *ZmGa1Ps-m*, *ZmPME3* and *ZmPRP3* constitute the UCI system in maize

To further investigate the genetic relationship between silk-expressed *ZmPRP3* and *ZmPME3* and how they work together to regulate *ga1* pollen tube growth, we observed Z58 pollen tube growth rates in silks of three recombinants: R2, R3, and R5 (Fig. 1b). Quantitative RT-PCR analysis confirmed that the expression levels of *ZmPME3* in silks were reduced by about half in heterozygous lines compared with homozygous lines for the SK genotype at Component1. *ZmPRP3* is only expressed in lines in which Component2 harbors the Z58 genotype (Fig. 3a). At 8 h after pollination, Z58 pollen tubes grew longer in silks of RL-Component1^Heterozygous^-Component2^Z58^ (R^H-Z58^) plants than in the RL-Component1^Heterozygous^-Component2^SK^ (R^H-SK^) plants [3.99 ± 0.43 cm vs. 2.49 ± 0.38 cm, ($P = 2.29\text{E}-22$)] (Fig. 3b). However, they showed a comparable growth rate in silks of RL-Component1^SK^-Component2^SK^ (R^SK-SK^) and RL-Component1^SK^-Component2^Z58^ (R^SK-Z58^) [2.07 ± 0.28 cm vs. 2.06 ± 0.26 cm, ($P = 0.93$)] (Fig. 3b). These results illustrate that the effect of Component2 on regulating pollen tube growth depends on Component1 genotype, is only relevant when Component1 is heterozygous. Component2^Z58^ promotes Z58 pollen tube growth. This observation explains the lack of segregation distortion in R3 (Fig. 1b).

To test whether the relationship between the two functional components depends on *ZmPRP3* and *ZmPME3*, we crossed *CR-Zmprp3* cas9-positive plants with *ZmPME3* over-expression plants. In the resultant hybrid plants, *ZmPRP3* was edited and generated a 147 bp deletion (*ZmPME3^OE+^/ CR-Zmprp3*) (Figs. 2g and 3c). At 8 h after pollination, we observed that the length of KN5585 (*ga1*) pollen tubes went from longest to shortest as follows: in WT [KN5585 (*ga1*)] silks (4.18 ± 0.23 cm) > in *CR-Zmprp3* (2.16 ± 0.29 cm) or in *ZmPME3^OE+^* (2.04 ± 0.42 cm) silks > in *ZmPME3^OE+^/ CR-Zmprp3* silks (1.22 ± 0.23 cm; 1.35 ± 0.20 cm) > in SK silks (1.15 ± 0.29 cm) (Fig. 3d). Furthermore, the seed set ratios in crosses between *ZmPME3^OE+^/CR-Zmprp3* and WT were significantly reduced and comparable to that of crosses between SK and WT. (Fig. 3e−g and Supplementary Fig. 20).

Collectively, our results revealed that three types of genes possess both male and female determinant functions, independently regulate

the growth rate of pollen tubes and thus affect cross incompatibility in maize. Based on the classic genetic model of the *Ga1* locus[32], we are now ultimately able to demonstrate causal genes governing the UCI system that balance antagonistic and synergistic effects. *ZmGa1Ps-m* genes are male determinants that are capable to endow *Ga1-S* and *Ga1-M* pollen with compatibility in any type of silks. Silk-expressed *ZmPME3* is a female determinant that blocks *ga1* pollen tube growth. On this basis, the silk-expressed gene *ZmPRP3* that exists in *ga1* and *Ga1-M* silks acts as an accelerator of pollen tube growth. Knocking out *ZmPRP3* and over-expressing *ZmPME3* produces the greatest effect on blocking *ga1* pollen tube. Introducing *ZmPRP3* in the heterozygous *ZmPME3* background can effectively break the block effect of *ZmPME3*, promote *ga1* pollen tube growth (Fig. 3h−k and Supplementary Fig. 27). The presence of *ZmPRP3* may explain why a few *Ga1-M* silks with intact *ZmPME3* gene and *ZmPME3^OE+^* plants can still be fertilized by *ga1* pollen as reported also in the previous study[15].

## The *Ga1* locus diverged before maize domestication and was further modified during maize improvement

We next attempted to study the evolutionary origin of the *Ga1* locus. For this purpose, we performed a principal component analysis (PCA) with 130,680 combined SNPs (53,173 SNPs derived from the B73 reference genome and 77,507 derived from the SK reference genome) around the *Ga1* locus on 845 maize germplasms [392 diverse maize inbred lines, 336 landraces and 117 wild maize [*Z.mays ssp parviglumis* ($n = 68$), hereafter named *parviglumis*; *Zea mays ssp. mexicana* ($n = 49$), hereafter named *mexicana*][43]. PC1 clearly separated *parviglumis* and *mexicana* from maize (landrace, *ga1*, *Ga1-M*, *Ga1-S* type lines), PC2 separated *parviglumis* into two sub-clusters [*par1* ($n = 19$) and *par2* ($n = 49$)] (Fig. 4a), as well as landraces [*lan1* ($n = 236$) and *lan2* ($n = 100$)] and modern maize [*ga1* ($n = 303$) and *Ga1-S/Ga1-M* ($n = 7/82$)] (Fig. 4a). Notably, *mexicana* is most closely related to *par2*, *lan1* contained *ga1* type lines, and *lan2* contained *Ga1-M* and *Ga1-S* type lines. We next genotyped *ZmGa1Ps-m* and *ZmPRP3* genes across all 820 lines by calculating mapped reads count, and found that *ZmPRP3* was present in all lines except *Ga1-S* type lines (Supplementary Fig. 28 and Supplementary Data 6). *ZmGa1Ps-m* genes were present in both *Ga1-M* and *Ga1-S*, but not in *ga1* type lines. Moreover, the mapped reads count of *ZmGa1Ps-m* genes in *par1* were significantly higher than in *par2* (Supplementary Data 6). Similar trends also appeared in the landrace (Supplementary Data 6) where *ZmGa1Ps-m* genes were almost absent in *lan1* but displayed enriched mapped reads count in *lan2* (Fig. 4b and Supplementary Fig. 28). The haplotype analysis on *ZmPME3* further showed that *par1*, *lan1* and *ga1* displayed a very consistent haplotype, while a proportion of *par2*, *mexicana* and *lan2* possessing *ZmPME3* haplotype that were more similar to *Ga1-S* and *Ga1-M* type lines (Supplementary Fig. 29). These results suggest that the origin of *Ga1-S* and *Ga1-M* type lines are close and related to *par2*, *mexicana* and *lan2*.

Altogether, these results strongly indicated that the divergence of the *Ga1* locus occurred already before maize was domesticated from teosinte, divergence continued during further maize improvement. One evolutionary branch (*par1−lan1−ga1*) maintains the same haplotype across three types of genes, which is a combination of low copy number of *ZmGa1Ps-m*, non-functional *ZmPME3* (barely expressed and lack of complete gene sequence) and *ZmPRP3* [*ZmGa1Ps-m-*, *ZmPME3-*, *ZmPRP3* + (−+)]. The second evolutionary branch: *par2* and *mexicana* contains high copy numbers of five pollen expressed *PME* gens, *ZmPME3* with potential enzymatic activity, and *ZmPRP3* [*ZmGa1Ps-m* +, *ZmPME3* +, *ZmPRP3* + (+++)], which was domesticated to *lan2* [*ZmGa1Ps-m* +, *ZmPME3* +, *ZmPRP3* + (+++)]. An improved population was selected from *lan2* proceeded in two directions: one is the loss of *ZmPRP3*, forming modern *Ga1-S* type lines [*ZmGa1Ps-m* +, *ZmPME3* +, *ZmPRP3-* (++−)], the other is the loss of *ZmPME3* function (caused by a 1 or 2 bp insertion, which is different from the haplotype in *ga1* lines), forming modern *Ga1-M* type lines [*ZmGa1Ps-m* +, *ZmPME3-*,

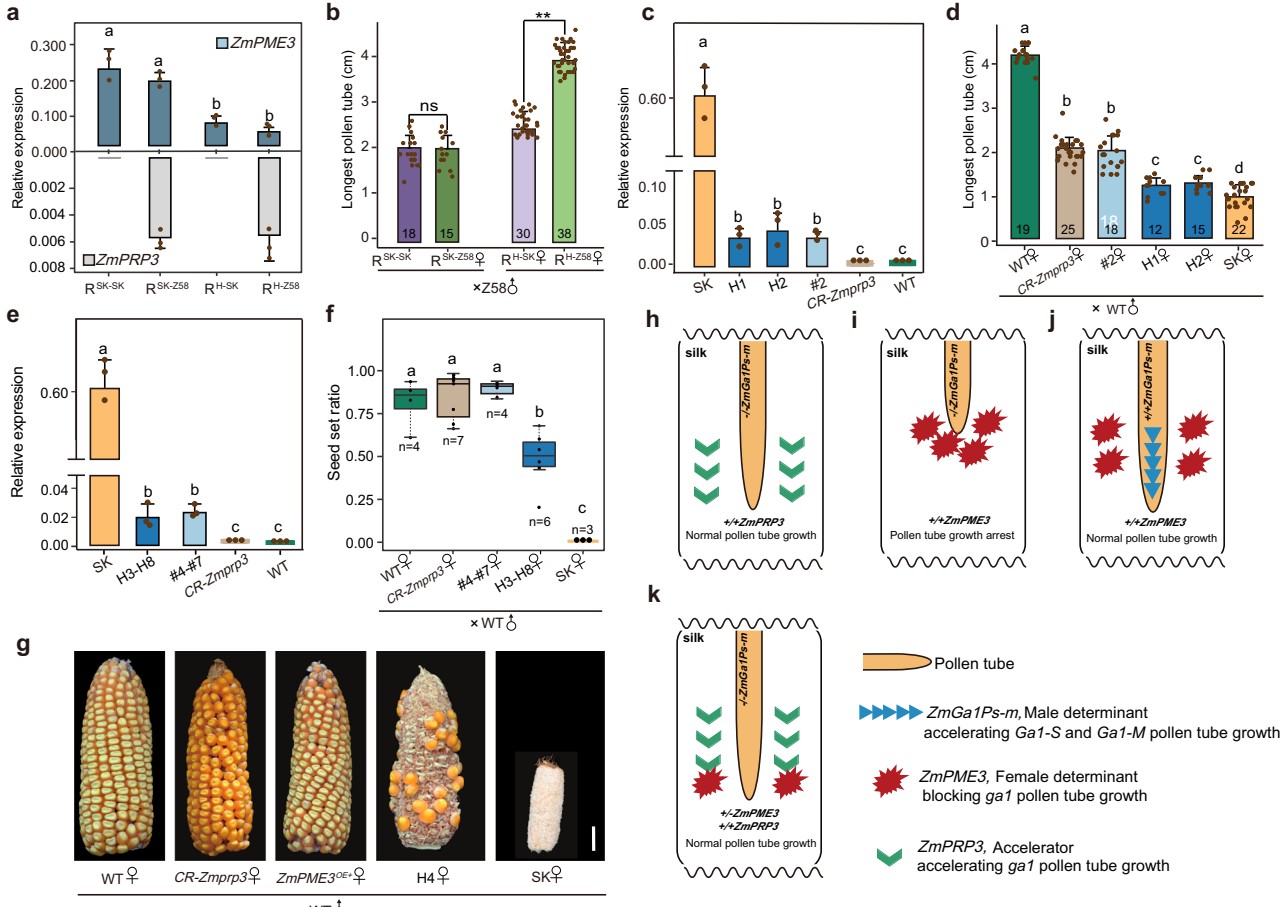

**Fig. 3 | Genetic relationship between three types of genes of the _Ga1_ locus.**
**a** Quantitative expression analysis of _ZmPRP3_ and _ZmPME3_ in silks of R[SK-Z58], R[SK-SK], R[H-Z58], and R[H-SK]. **b** Comparison of the longest pollen tubes of Z58 at 8 h after pollination in silks of R[SK-Z58] and R[SK-SK] silks, and in R[H-Z58] and R[H-SK]. **c** Quantitative expression analysis of _ZmPME3_ in silks of WT, _ZmPME3_[OE+], _CR-Zmprp3_, and _ZmPME3_[OE+]/ _CR-Zmprp3_ plants. #2 represents _ZmPME3_[OE+] plant. H1 and H2 represent two _ZmPME3_[OE+]/ _CR-Zmprp3_ plants. **d** Comparison of the longest pollen tubes of WT [KN5585 (_ga1_)] at 8 h after pollination in silks of _CR-Zmprp3_, _ZmPME3_[OE+], WT, SK and _ZmPME3_[OE+]/CR-_Zmprp3_ plants. **e** Quantitative expression analysis of _ZmPME3_ in silks of _CR-Zmprp3_, _ZmPME3_[OE+], WT, SK and _ZmPME3_[OE+]/CR-_Zmprp3_ plants. H3-H8 represents six _ZmPME3_[OE+]/CR-_Zmprp3_ plants. #4-#7 represents four _ZmPME3_[OE+] plant. **f** Statistical analysis of the seed set ratio from crosses by using SK, _ZmPME3_[OE+], _CR-Zmagp1_, _ZmPME3_[OE+]/CR-_Zmagp1_ and WT each as a female parent, WT [KN5585 (_ga1_)] as a common male parent. H3-H8 represents six _ZmPME3_[OE+]/CR-_Zmprp3_ plants. #4-#7 represent four _ZmPME3_[OE+] plants. **g** Crossing experiments showing ears of WT after self-pollination and crosses when SK, _ZmPME3_[OE+], _CR-Zmagp1_, _ZmPME3_[OE+]/CR-

_Zmagp1_ were used as female parents, WT [(KN5585, _ga1_)] pollen was used as male parent. Scale bar = 2 cm. **h** _ga1_ and _Ga1-M_ silk-expressed _ZmPRP3_ acts as an accelerator, promoting the growth of _ga1_ pollen tubes. **i** Silk-expressed _ZmPME3_ exists in _Ga1-S_ silk and is a female determinant that blocks _ga1_ pollen tube grow through silk. **j** _ZmGa1Ps-m_ genes are male determinants, which exist in _Ga1-S_ and _Ga1-M_ pollen and can completely overcome block effect of _ZmPME3_. **k** When _ZmPME3_ is heterozygous, introducing _ZmPRP3_ can effectively break the block effect of _ZmPME3_, promote _ga1_ pollen tube growth. For **a**, **c**–**f**, data were analyzed according to the LSD test. For **b**, data were analyzed by two-tailed Student's _t_-test. Error bars represent mean + SD (**a**–**e**). a, b, c indicate significant differences (_P_ < 0.01) (**a**, **c**–**f**). Asterisks indicate significant differences (**_P_ < 0.01) (**b**). ns (not significant, _P_ > 0.05) (**b**). Relative expression (_n_ = 3, biologically independent samples) (**a**, **c**, **e**). Longest pollen tube length (_n_ = number of biologically independent samples in each column) (**b**, **d**). In box plots, the center line represents the median, box edges delimit lower and upper quartiles and whiskers show the highest and lowest data points, _n_ present sample size (**f**). Source data are provided as a Source Data file.

_ZmPRP3_ + (+−+)] (Fig. 4c and Supplementary Fig. 30). Given the huge structural variation at the _Ga1_ locus, the occurrence of recombination was inhibited and thus the independent variation of the three type of maize was maintained.

## Discussion

Unilateral cross incompatibility (UCI) is a fascinating biological phenomenon, related to prezygotic speciation processes, and can offer an effective method of manipulating crosses for crop improvement. Although it was discovered nearly a century ago, we are only now beginning to understand the genetic model and molecular mechanisms underlying this phenomenon. Here, we reported three different types, a total of seven linked genes underlying the _Ga1_ locus, which control UCI phenotype by affecting pollen tube growth with both antagonistic and synergistic manner. After artificially creating three

haplotypes that do not exist in nature [_ZmPME3_[OE+]: _ZmPME3_ + , _ZmPRP3_ + (++); _CR-Zmprp3_: _ZmPME3_-, _ZmPRP3_- (−)], we can now complete a total of eight cross combinations between the three types of genes (Fig. 4d). The observed phenotypes of these cross combinations matched with the expected phenotype from genotypes across three types of genes. According to previous studies, _ga1_ pollen tubes accumulate excess levels of highly methylesterified pectins at the apex region when growing through the _Ga1-S_ silk. However, this phenotype does not appear when _Ga1-S_ pollen tubes grow through the _Ga1-S_ silk or _ga1_ pollen tubes grow through the _ga1_ silk[14]. This suggest that silk-expressed and pollen-expressed _PME_ genes work together to maintain the balance of pectin esterification/de-esterification at the apical region of the pollen tube cell wall. Our current study provided direct evidence for the existence of two types of _PME_ genes and showed that over-expressed _ZmPME3_ in the _ga1_ silk can significantly decrease the

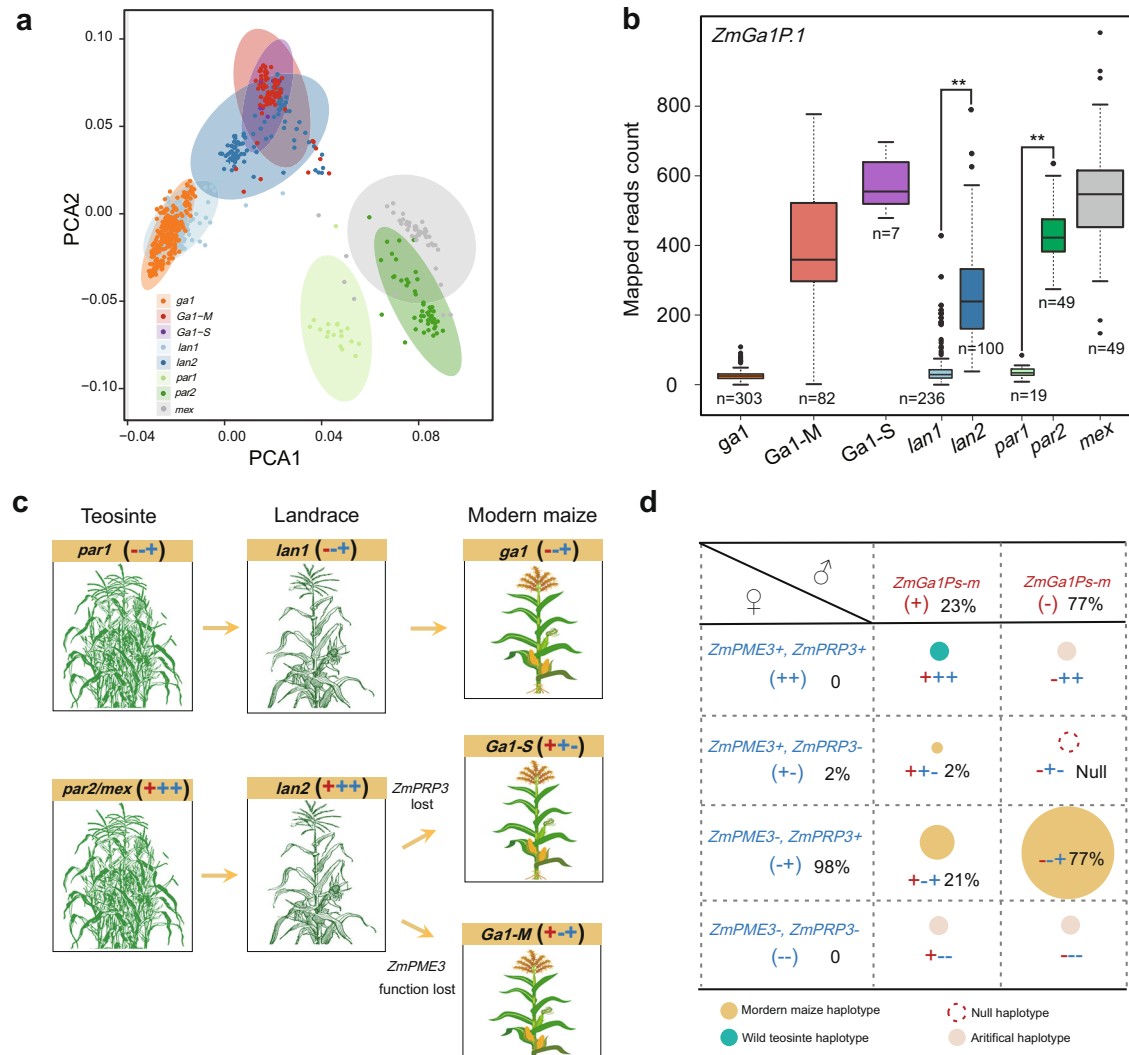

**Fig. 4 | Evolution and modification of the *Ga1* locus during maize domestication. a** Principal component analysis (PCA) of the *Ga1* locus in *parvivglumis*, *mexicana* (*mex*), landraces, *Ga1-M*, *Ga1-S* and *ga1* inbred lines. **b** Mapped reads count of *ZmGa1P.1* in the two *parviglumis* subgroups (*par1*, *par2*), *mexicana* (*mex*), two landrace subgroups (*lan1*, *lan2*) and modern maize subgroups (*Ga1-S*, *Ga1-M* and *ga1* inbred lines). **P < 0.01 (two-tailed Student's *t*-test and Wilcoxon ranked sum test). In box plots, the center line represents the median, box edges delimit lower and upper quartiles and whiskers show the highest and lowest data points. *n* present sample size. Mapped reads counts are provided in Supplementary Data 6. **c** A putative evolutionary history of the *Ga1* locus during domestication and improvement. The genotype of *ZmGa1Ps-m* is highlight with red, the genotype of *ZmPME3* and *ZmPRP3* are highlight with blue. −+ presents a haplotype that lost *ZmGa1Ps-m* and *ZmPME3* (lack of complete gene sequences in the genome) but possesses *ZmPRP3*. +++ presents a haplotype that possesses *ZmGa1Ps-m*, *ZmPME3* (with potential enzymatic activity) and *ZmPRP3*. ++− presents a haplotype possesses *ZmGa1Ps-m*, functional *ZmPME3* but lost *ZmPRP3*. +−+ presents a haplotype that

possesses *ZmGa1Ps-m*, *ZmPRP3* but *ZmPME3* lost their funtion (due to 1 or 2 bp insertion). **d** Eight cross combinations across three types of genes. *ZmGa1Ps-m* genes are highly expressed in pollen, the genotype of *ZmGa1Ps-m* is highlight with red. *ZmPME3* and *ZmPRP3* are expressed in silk, the genotype of *ZmPME3* and *ZmPRP3* are highlight with blue. Three haplotypes exist in modern maize inbred lines [*ga1*:−+; *Ga1-S*: + + −; *Ga1-M*: + − + ]. One haplotype exists in wild teosinte [*par2/Mexicana*: +++]. Null haplotype presents cross incompatibility between *ga1* pollen and *Ga1-S* silk [*ZmGa1Ps-m-, ZmPME3 + , ZmPRP3-* (−+−)]. Three artificial haplotypes that do not exist in natural collections were: *ga1* pollen crossed onto *ZmPME3* over expressed lines [*ZmGa1Ps-m-, ZmPME3 + , ZmPRP3 + (−++)] Ga1-S* pollen crossed onto *ZmPRP3* knock-out lines [*ZmGa1Ps-m + , ZmPME3-, ZmPRP3- (+−)], and *ga1* pollen crossed onto *ZmPRP3* knock-out lines [*ZmGa1Ps-m-, ZmPME3-, ZmPRP3- (−−−)], which all resulted in successful crosses, proving the three artificial haplotypes are fertile. The number labeled in black represent the proportion of male gametes, female gametes and zygote haplotype in AMP.

degree of methylesterification of silk cell walls and decrease *ga1* pollen tube growth rate, providing valuable information about how *PME* genes regulate UCI. It is possible that (i) *ZmPME3* activity accumulates more low-methylesterified pectin, therefore solidifies the silk cell walls and thus establishes a stronger barrier for growing pollen tubes[44]; (ii) *ZmGa1Ps-m* inactives *ZmPME3*, thereby overcoming the incompatibility effect of *ZmPME3*.

*ZmPRP3* appears to be an accelerator of pollen tube growth in a pathway independent from *PME* genes, and thereby plays an auxiliary role in the *Ga1* genetic model. To date, secreted *CRPs* have been mainly

reported as ligands of receptors or other cell surface proteins, such as *LUREs* and *RALFs* in *Arabidopsis*[45–48], and *ES1-4* in maize[49]. It thus appears very likely that *ZmPRP3* affects pollen tube growth after ligand-receptor binding. The fact that enriched DEGs in maize lines lacking *ZmPRP3* were related to metal ion binding suggest that pathways controlled by *ZmPRP3* display a high redox potential and may involve ROS metabolism[50,51]. Future studies are required to investigate whether a receptor exists in the *ga1* pollen tube that interacts with *ZmPRP3*, in order to determine the mechanism underlying its pollen tube growth promoting activity.

In summary, this study provides important scientific insights into the genetic and molecular basis of UCI and demonstrates its clear evolutionary route, propelling us forward in better understanding reproductive isolation and evolutionary history of modern maize. This knowledge can be used to enhance maize improvement, such as creating artificially isolated varieties for the preservation of important germplasm resources and preventing gene flow from transgenic plants to existing core germplasms in future breeding strategies.

## Methods

### Fine mapping of the *Ga1* locus

A high-density genetic map for the Z58 × SK recombination inbred lines (RIL) population was constructed with 2798 genetic bins[21] via the Illumina MaizeSNP50 BeadChip (Illumina Inc) that contains 56,110 single nucleotide polymorphisms (SNPs), and thus allows to scan the genotype ratio of the two parents in each RIL on the whole genome level. On the basis of Mendel's model, the expected ratio of the two parents genotype is 1:1. We calculated the actual genotype ratio of for each bin and used the Chi-square Test to assess segregation distortion locus by setting $P = 1.79E-06$ ($P \le 0.05/2798$) as the threshold value for segregation distortion significance.

A F$_2$ mapping population was produced from a cross between SK and Z58, approximately 4900 individuals resulted from the self-pollination of the F$_2$ plant and were genotyped with molecular markers (see Supplementary Table 2) to identify the degree of segregation distortion using the Chi-square test. Molecular marker M1 is cited from Bloom and Holland[24]. M5 and M6 are cited from ref. [22]. Segregation distortion was used as phenotype to narrow down the *Ga1* candidate region.

### Genome-wide association analysis

A genome-wide association study (GWAS) of male function of the UCI system was performed using the mixed linear model[52] implemented in the software TASSEL 3.0[53] that took into account population structure (Q) and familial relationship (K). SNPs with minor allele frequency (MAF $\ge$ 5%) in the 498 inbred lines from association mapping panel (AMP) were employed in the association analysis. $P$-value of each SNP were calculated, and significance was defined at a uniform threshold of $8 \times 10^{-9}$ ($P = 0.01/N$, $N = 1,250,000$, Bonferroni correction). The phenotypic value of each inbred lines in the association test was assigned on a 0 to 3 scale (Supplementary Fig. 4) to present the degree of cross incompatibility and compatibility when CML304 (*Ga1-S*) was used as female parent and the inbred lines from the AMP were used as male parent.

### Identification of *Ga1-S, Ga1-M* and *ga1* inbred lines in the AMP

CML304 (*Ga1-S*) was used as female parent and inbred lines from the AMP were used as male parents to perform crossing experiments, and the ratio of seed set was used as phenotype to identify inbred lines' genotypes. When the ratio of seed set was less than 0.1, we identified inbred lines as *ga1* types. When Z58 (*ga1*) was used as male parent, *non-ga1* inbred lines were used as female parent, and the ratio of seed set was less than 0.1, we identified inbred lines as *Ga1-S* types. When the ratio of seed set was greater than 0.5, we identified inbred lines as *Ga1-M* types.

### SNP calling and principal component analysis

Deep DNA re-sequencing data (-20×) of all 367 lines from the AMP, 68 lines of *Zea mays ssp parviglumis* (the precursor of modern maize, hereafter named *parviglumis*) and 49 lines of *Zea mays ssp. mexicana* (hereafter named *mexicana*], and 336 lines of landraces were first processed using FastQC (v0.11.3) and Trimmomatic (v0.33) to remove poor-quality base calls and adaptors. Reads were then mapped to the B73 and SK reference genome using

bowtie2-2.3.3 (with the parameters "–very-fast and–end-to-end", respectively)[54]. Unique mapped reads were sorted and indexed using Picard (v1.119). SAMtools (v1.3.1)[55] and GATK (v3.5) were used for SNP calling. SNP call was carried out with mapping quality (MQ > 20.0), and thresholds set by sequencing coverage based on minimum coverage (DP > 5) and maximum coverage (DP < 200). Variants from 820 lines were then combined by GATK Combine-Variants into a single variant calling file. Finally, sites with a missing rate lower than 25% in all samples were included. Smartpca[56,57] was used to perform principle component analysis (PCA) with 130,680 combined SNPs (53,173 SNPs from the B73 reference genome, and 77,507 from the SK reference genome).

### Calculating reads count, visualization of haplotype clusters

Considering the high sequence identity shared among *ZmGa1Ps-m* genes, we used all reads without filtering unique reads. Read counts of each gene within the *Ga1* locus were calculated using subread-1.6.3 tools featureCounts[58]. 141 SNPs (33 SNPs from a 1Kb upstream of *ZmPME3*, 58 SNPs within *ZmPME3* gene region and 50 SNPs from a 1 Kb downstream of *ZmPME3*) were selected. Haplostrips[59] was used to reveal haplotype clusters in multiple populations.

### Genomic variation detection

A genomic fragment of B73 v4 from chromosome 4: 7 Mb - 12 Mb was used as the initial reference genome, a SK genomic fragment derived from chromosome 4: 7 Mb -12 Mb was aligned to the reference genome with local NCBI-BLAST-2.7.1[60] following the parameters "-evalue 1e-5 -max_target_seqs 1 -num_threads 10 -perc_identity 95". 1 kb sequence upstream and downstream of each annotated gene within the *Ga1* locus based on SK and B73 genome was collected and aligned to the whole B73 and SK genomes using parameters "-evalue 0.01 -max_target_seqs 1 -num_threads 10 -perc_identity 70" in order to identify homologs. Genomic sequence alignment was displayed using the R package genoplotR[61], and homologous gene alignment was generated using jcvi[62].

### RNA-seq analysis

Total RNAs were extracted from pollen and silks of SK, P178 and Z58 using the Quick RNA Isolation Kit (Huayueyang, China). Three technical replicates were performed for each sample. RNA libraries were constructed according to the TruSeq1 Stranded mRNA Sample Preparation Guide (TruSeq1 Stranded mRNA LTSetA). All libraries were PE150 sequenced using an Illumina HiSeq3000 system. Low quality reads were filtered out with Trimmomatic (v0.36)[63]. RSEM-1.3.0[64] was used to align RNA-seq reads to the B73 v4 and SK reference genome respectively, and gene expression levels were estimated using default parameters.

### Quantitative RT-PCR and re-sequencing of *ZmPME3* and *ZmPRP3*

Total RNAs were extracted from silks of 70 accessions from the AMP and silks or pollen from transgenic maize plants using the Trizol reagent (Invitrogen). First-strand cDNA was synthesized using EasyScript One-Step gDNA Removal and cDNA Synthesis SuperMix (Transgene Biotech). Real-time fluorescence quantitative PCR with SYBR Green Master Mix (Vazyme Biotech) on a CFX96 Real-Time System was performed to quantify relative expression levels of *ZmPRP3* and *ZmPME3*. Each set of experiments was repeated three times. The relative quantification method ($2^{-\Delta\Delta Ct}$) was used to evaluate quantitative variation. Primers used to re-sequence and quantify expression level of *ZmPME3* (in 70 maize inbred lines[15]) listed in Supplementary Table 3. Primers that used to amplify and sequence gene regions of *ZmPRP3* (in 38 maize inbred lines) are also listed in Supplementary Table 3. Sequences were aligned with mafft-7.037[65,66] to identify small indels and other variants in the different maize lines.

## Transgenic validation of *ZmPRP3*, *ZmGa1Ps-m* genes, and *ZmPME3*

We synthesized a 1261 bp genomic DNA fragment from *ZmGa1Ps-m* genes (from ATG to TAA, ref to SK) and 1296 bp genomic DNA fragment from *ZmPME3* (from ATG to TGA, ref to SK) for the over-expression study. The *ZmUbi* promoter was inserted into the modified binary vector pCAMBIA3300 to drive gene expression and transformed into the maize inbred line KN5585. We also performed CRISPR-Cas9 based gene editing for four *CRP* genes. Two *ZmPRP3* specific guide-RNAs were incorporated into the *pCPB-ZmUbi-hspCas9* vector[39] and targeted *ZmPRP3* single exon. Two *Zm00001d048948* specific guide-RNAs and two guide-RNAs for targeting *Zm00001d048949* and *Zm00001d048950* were also incorporated into *pCPB-ZmUbi-hspCas9* vectors at once to generate *CR-Zm00001d048948* mutant and *CR-Zm00001d048949*, *CR-Zm00001d048950* double mutant respectively. Primers and gene sequence used for transgenic experiments are listed in Supplementary Table 3.

## In vivo pollen-tube growth assay

All experimental silks were collected at 8 or 24 h after pollination, and fixed in Carnoy's fluid for 12 h. Fixed silks were washed in a series with the 30%, 50%, 70% cethanol and immersed in 2 M NaOH overnight for tissue softening. Silks were then stained with 0.1% aniline blue (Amresco) for 2 h. Samples were observed under a Nikon ECLIPSE 90i microscope equipped with an ultraviolet filter. Fifteen pollen tubes from at least three plants were examined in each cross.

## Immunolabeling analysis

To visualize the degree of pectin esterification in different silk cells at 8 h after pollination, silks were first fixed in 4% FAA. After dehydration using a 30%, 50%, 70% ethanol series, silks were embedded in paraffin. 2 μm sections were cut from embedded samples and collected on glass slides, and incubated with primary antibody (LM20; LM19, Plant Probes, diluted at 1:20) in a PBS-Tween buffer (1% BSA and 0.05% Tween 20) for 1 h at room temperature. FITC coupled anti-rat IgG (ABclone) was used as a secondary antibody at a 1:500 dilution. Fluorescence signals were recorded with a confocal laser scanning microscope (Zeiss) and measured with ImageJ software[67]. Each image was transformed to an eight-bit image, Next, signal areas were selected with the option "create selection" and measured with the menu tool "measure". Signal intensity was measured and represented as total fluorescence value divided by the area. 15 silk cells from at least three individual plants were employed for analysis.

## Statistical analysis

Statistical calculations (mean, SD, *t*-test, LSD test and Wilcoxon ranked sum test) were performed by using Microsoft Excel software and R. Shapiro–Wilk normality test was used to test the normal distribution of each set of data (Supplementary Data 7). Statistical significance of differences between sets of data was assessed by using a two-tailed Student's *t*-test and Wilcoxon ranked sum non-parametric test. The statistical significance of differences between multiple groups of data was determined by using a LSD test. Differences between two datasets or multiple datasets were considered significant at $P < 0.01$. Exact $P$ values are provided in Source Data file.

## Reporting summary

Further information on research design is available in the Nature Research Reporting Summary linked to this article.

## Data availability

RNA-seq sequence data from this study can be found under BioProject accession PRJNA778568. Re-sequence data of the 336 maize landrace can be found under BioProject accession PRJNA783885. The previously reported SK genome sequence can be found in MaizeGDB [https://download.maizegdb.org/Zm-SK-REFERENCE-YAN-1.0/]. Source data are provided with this paper.

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

## Acknowledgements

We greatly appreciate Dr. Xianran Li from Iowa State University, Dr. Lele Wang from University of Regensburg and Mr. Luo Cheng for providing

the critical comments for the manuscript. This research was supported by the National Key Research and Development Program of China (2020YFE0202300) and the National Natural Science Foundation of China (31961133002 and U1901201).

## Author contributions

J.Y. conceived and supervised this study. Y.W., W.L., L.W., and C.W. developed the mapping populations, managed the field work and prepared the samples. Y.W., S.G., and N.Y. performed the phylogenetic analysis and genomic variation analysis. Y.W., W.L., and G.C. finished the fine mapping and GWAS. Y.W., W.L., Jiali Y., Yuqing W., S.L., and J.X. finished the transgenic experiment. Y.W. finished the RNA-seq data analysis, the observation of pollen tube growth and immunolabeling analysis. Y.W. and Jiali Y. finished sequencing work. Y.W, G.L., T.G., Y.X., M.W., A.F., T.D., and J.Y. discussed the data and prepared the manuscript. All authors read and approved the manuscript.

## Competing interests

The authors declare no competing interests.
