## [Peer Review File · Nature Communications]

Three types of genes underlying the Gametophyte factor1 locus cause unilateral cross incompatibility in maizeReviewers' Comments:

Reviewer #1:

Remarks to the Author:

Key Results: The most noteworthy result of this manuscript is the discovery of a novel function for the pathogenesis-related protein3 (*prp3*) gene on chromosome 4. This new role is in pollination. Its presence in the silk blocks the ability of *Ga1s* to function in pollen exclusion, and its absence is required for *Ga1s* gametophytic incompatibility. This discovery allowed the authors to confirm the role of *ZmPme3* as the female component of *Ga1s*, which has eluded other researchers. They also make an interesting attempt to understand the evolution of the locus.

These two discoveries are of significance to the field and the data is worthy of publication, but major revisions would need to be done. There are other results and observations within the manuscript which have already been reached in previous publications. All of the relevant publications are cited in the manuscript, but the authors do not make clear that their results are confirming published observations, starting with omitting the discovery of *ZmPme3* in line 74. Specific instances of omission will be noted below.

The renaming of *prp3* to *Zea mays Accelerator of Pollen tube Growth1 (ZmAPG1)* should be only done according to the standard of Maize Genetics Nomenclature (Maize Genetics Nomenclature (maizegdb.org). As the name APG does not encompass all the functions that *prp3* is known to have, care should be taken. They would also need to acknowledge that they are changing a gene name.

A related problem is the extraneous use of a Chinese legend to make a model of the *Ga1s* locus function (Fig. 3h). A simple model using biologically relevant descriptions would be preferred. A model has already been described for maize UCI (reference below), and the authors can indicate how they have improved this model.

The GWAS results presented in this study are a minor result, but I am surprised that they only detect signals on chromosome 4 (Sup Fig 4b). Hurst et al., 2019 detected 7 traits on five chromosomes. The report of multiple PME (*ZmPmes-m*) in addition to *Ga1P* (or instead of?) in the pollen of *Ga1s* is inconsistent with the published literature (and unpublished data I have). The authors do not adequately acknowledge this discrepancy and as such, do not adequately show that these are unique functional genes rather than pseudogenes which have been previously reported. It seems more that they have identified variants of the *Ga1P* found in SDGa25 (and Hp301). The transgenics seem to be a good experiment, but the results are not a very strong case, especially when considering that transgenic *Ga1P* pollen leads to full seed set on *Ga1s* silks. In fact, I'd say that they almost show that the *ZmPmes-m* variants aren't good candidates for the male factor. They only indicate one primer pair for the qPCR, so there's no independent confirmation that multiple transcripts are present (leading to unique proteins), as the primers would pick up all. Perhaps, their data indicates copy number variation in the SK line? It's one thing to show you can make a protein that has some function, it's another to show that the protein is there normally. Hp301 has an available genome sequence to make comparisons and strengthen the conclusions. Proteomic evidence would also go a long way in verifying multiple male PMEs from the *Ga1* locus, as proteomic data from SDGa25 does not show additional *Ga1P* variants in the pollen.

There are several instances of grammatical errors, but not worth mentioning as a major rewrite is in order. The references 1-45 are separated from the remainder 46-61 by methods, acknowledgments, etc. Line 119: this is the first report that I can find of SK and CML304 being *Ga1s*. If not, please reference.

Validity: The methods are sound, although the interpretations are sometimes questionable or identical to previous work and not properly credited.

Most of the work can be reproduced, however, the genome sequence for SK cannot be accessed in the U.S. At least the link does not work from their Yang 2019 publication.

Specific instances where results are the same as published reports:

- Pollen tube measurement conclusions were the same as discussed in Zhang, et al 2012 (their reference 8). The authors chose later time points but came to the same conclusion without acknowledging that. This was also reported (and adequately acknowledged) in Moran Lauter, et al. 2017.
- Line 102: Segregation distortion at this locus has been observed since the 1920s and reported many times. The authors' contribution is separating out component2, and this needs to be made clear.
- Fig 1. ZmPme3 gene structure in Ga1s and B73 (28. Moran Lauter, et al. 2017); ZmPMEs-m shown in (14) Zhang, et al. 2018 as pseudogenes.
- Insertion in ZmPme3 in the Ga1m line NC390xNC394 reported in 28. Moran Lauter, et al. 2017
- Line 155: Fine-mapping has been successful yielding the same interval shown here, and it should be cited (Bloom and Holland, 2011; 8. Zhang et al.,2012, 21. Liu et al., 2014) Some of the primers used in this study are identical to those used in the Liu 2014 study.
- Line 179: it isn't made clear that Zm00001d048936 is the non-functional B73 version of Ga1P that is reported in 14. Zhang et al, 2018.
- Line 191: 14. Zhang et al., 2018 showed a different variation between B73 and Ga1P. This should be noted.
- Lines 210-218: all reported in 28. Moran Lauter, et al. 2017 and not cited.
- qPCR and Resequencing primers are from 28. Moran Lauter, et al. 2017 (therefore, they had to know that they were repeating this work)
- Supplemental Fig 12 is reported in 29. Lu, et al. 2019 (Sup Fig 7)
- Model in 3h is a variation on Lu, et al. Plant Reproduction 33, pages 117–128 (2020) (paper not cited in the manuscript)

My summary: The main discovery in this work is a new and important role for prp3 in the regulation of pollination, and its ability to block ZmPme3 action in Ga1s gametophytic incompatibility. This discovery allows the authors to confirm that ZmPme3 is the causal silk factor in Ga1s UCI. Their data suggests that some previously reported Ga1P copies are actually transcribed but don't quite show a functional role. There are several instances of data confirming published results, which do not add knowledge to the field.

Reviewer #2:

Remarks to the Author:

Wang et al., perform a genetic analysis to uncover the genetic determinants regulating unilateral cross compatibility in maize in the male and female. As a result of their extensive genetic analysis the authors identify a group of pollen expressed PMEs and a silk-expressed PME3 and cysteine-rich protein. To validate the role of these genes in the control of UCI, the authors use CRISPR-cas9 and overexpression approaches to create different gene combinations with the aim to determine the function of the different candidate genes in the control of pollen tube growth. In addition, the authors also investigated the methylesterification status of cell walls of silk cells in different genetic backgrounds. Wang et al., in their study, they also investigate the evolutionary origin of the Ga1 locus.

The authors have done an extensive genetic analysis and used proper genetic validation tools to characterize their candidate genes. In their model the authors suggest that activation of PME3 may lead to the condensation of silk cell walls creating a physical barrier

Major :

a)The role of PMEs in the regulation of pollen dynamic and growth has been already described in other model species like Arabidopsis. I think the Introduction will benefit from including that with a few relevant references (as an example : Bosh et al., Plant Cell, 2005).

b) The authors check the esterification status of silk cells using LM20, and mainly binds to methyl esterified pectins. In their model they suggest that PME3 activation may lead to condensation of the cell wall establishing a physical barrier for growing pollen tubes. Have the authors checked with the 2F4 antibody (binds to Ca egg-box pectin) whether in the case of their PME3Ox lines the pattern is different compared to Zmapg1, Wt and SK genotypes ?

c) When the authors described that pollen tube stop growing. Do they pollen tubes burst ? or they just maintain their integrity but stop growing ?

d) It would be good that the authors would include the independent data points in their bar graphs, together with the error.

e) the authors should also check what type of data population they are analyzing. Most likely a t-test would not be the best tool. If that would be the case it should be described and justified in the methods.

Minor :

There are some grammar errors in the text. It would be good to revise the text gain. Here are two examples :

Line 100 : An segregation distortion locus that the genotypes of two parents significantly deviation from the expected ratio (1:1) was detected on chromosome 4 from 2Mb to 25Mb (Supplementary Fig. 2), which overlapped with the defined the Ga1 locus in previous reports

Line 518: ZmPMEs-m inactive (inactivates?) ZmPME3, thereby overcoming the incompatibility effect of ZmPME3.

Reviewer #3:

Remarks to the Author:

The main goal of the manuscript by Wang and collaborators is to identify the genetic determinants of the unilateral cross incompatibility (UCI) in the genus *Zea*. UCI has been observed both, in crosses between maize varieties and, in crosses between maize and their closest wild relatives, the teosintes. UCI acts as prezygotic reproductive isolation (RI). Dissecting mechanisms behind RI is interesting in several respects: it can illuminate the early set of reproductive barriers that may be selected to maintain the integrity of locally-adapted plants in face of maladaptive gene flow, it is also useful for the use of genetic resources in plant breeding which requires to overcome such barriers. I enjoyed reading this paper, it is well written and certainly is a great advance to our understanding of the complex genetic system underlying UCI. The authors fine-mapped the Ga1 locus using RIL populations; they recovered complete alignments of the region from the maize reference genome B73 (ga1 type that can accept any type of pollen) and SK (Ga1-S type that cannot be pollinated by ga1 pollen) revealing extensive structural variation including 13 consecutive PME genes in SK versus a single one in B73. The authors confirmed the role of genes encoding pectin methyltransferase which act as male determinants to restore pollen compatibility. They further showed that the Ga1 locus is composed of 7 expressed linked genes: five ZmPMEs-m genes expressed in the pollen, one silk-expressed ZmPME3 gene, and the newly-discovered ZmAPG1 gene. They used measures of seed set ratios from crosses involving the three haplotypes at the Ga1 locus (represented in an association mapping panel: ga1, Ga1-S, Ga1-M) and ga1 as male parent, RNA-seq, expression assays and transgenics to show that (1) Cumulative expression of ZmPMEs-m genes accelerate pollen tube

growth in Ga1-S and GA1-M pollen; (2) ZmPME3 is expressed in silks of all Ga1-S lines where it blocks ga1 pollen tube growth but is either not expressed or interrupted in ga1 or Ga1-M lines; ZmAPG1 gene is highly expressed in ga1 and Ga1-M silks but absent from Ga1-S lines, and likely accelerates pollen growth in the former, although it is not sufficient alone. ZmPMEs-m genes are capable to endow Ga1-S and Ga1-M pollen become compatibility in any type of silks. Hence, pollen tube growth is determined by a balance of pectin esterification-de-esterification at the apical region of the pollen tube controlled by PME genes, while ZmAPG1 plays an auxiliary role via an independent pathway. Finally, the authors have investigated the diversity around the Ga1 locus using 771 diverse germplasm including maize and parviglumis, and further sequenced the focal genes. They found that ZmAPG1 was absent from Ga1-S lines, and that ZmPMEs-m genes were absent from ga1 lines. All the analyses are well conducted and they support the conclusions.

I have a few comments that should help improving the manuscript.

(1) The evolutionary analysis would benefit from discussion extension about what we know about cross-incompatibilities between maize and teosintes. To my knowledge, such incompatibilities are particularly strong between maize and the subspecies mexicana. I would include mexicana in the sampling.

(2) Overall, the paper is quite complex and would benefit greatly from a figure that would summarize the results. In the current version, there is an attempt to do so in Figure 3h, but this figure could be improved and extended to present a broader overview of all the findings.

(3) A careful check of all figures would be good. For instance, in figure 1b, the authors should clarify what does the table stands for (segregation distortions), that H means Heterozygous etc.. The colours in Figure 1b stands for the genomic backgrounds, but in 1C the same colours are employed to designate gene-types, this can easily get confusing for the reader.

Point-by-point Response to Reviewer's comments:

Response to Reviewer#1:

Key Results: The most noteworthy result of this manuscript is the discovery of a novel function for the pathogenesis-related protein3 (PRP3) gene on chromosome 4. This new role is in pollination. Its presence in the silk blocks the ability of Ga1s to function in pollen exclusion, and its absence is required for Ga1s gametophytic incompatibility. This discovery allowed the authors to confirm the role of ZmPme3 as the female component of Ga1s, which has eluded other researchers. They also make an interesting attempt to understand the evolution of the locus.

These two discoveries are of significance to the field and the data is worthy of publication, but major revisions would need to be done. There are other results and observations within the manuscript which have already been reached in previous publications. All of the relevant publications are cited in the manuscript, but the authors do not make clear that their results are confirming published observations, starting with omitting the discovery of ZmPme3 in line 74. Specific instances of omission will be noted below.

My summary: The main discovery in this work is a new and important role for ZmPRP3 in the regulation of pollination, and its ability to block ZmPme3 action in Ga1s gametophytic incompatibility. This discovery allows the authors to confirm that ZmPme3 is the causal silk factor in Ga1s UCI. Their data suggests that some previously reported Ga1P copies are actually transcribed but don't quite show a functional role. There are several instances of data confirming published results, which do not add knowledge to the field.

Thank you for the comprehensive summary and praise. We very seriously considered all your criticism and have tried our best to cite all the related reported results at the right positions in the revised manuscript. We hope that the revised version can satisfy you.

Q1: The renaming of ZmPRP3 to *Zea mays* Accelerator of Pollen tube Growth1 (*prp3*) should be only done according to the standard of Maize Genetics Nomenclature (Maize Genetics Nomenclature (maizegdb.org)). As the name APG does not encompass all the functions that *prp3* is known to have, care should be taken. They would also need to acknowledge that they are changing a gene name.

[Response]: We greatly appreciate your thoughtful suggestion about the name of *Zm00001d048947*, we agree that the name APG does not encompass all the functions that this gene is known to have (*pathogenesis-related protein3 prp3*; Hawkins et al., 2018; **Ref. 1**), We noticed that the homologous gene of *Zm00001d048947* has also

been reported in *Arabidopsis*, called *PR4* (Mishina et al., 2007; Ref. 2), however, *ZmPR4* as a gene name has also been used by other unrelated genes in maize (Bravo et al., 2003; Ref. 3), so we decided use *ZmPRP3* as the gene name, and modified throughout the manuscript.

Q2: A related problem is the extraneous use of a Chinese legend to make a model of the *Ga1s* locus function (Fig. 3h). A simple model using biologically relevant descriptions would be preferred. A model has already been described for maize UCI (reference below), and the authors can indicate how they have improved this model.

[Response]: We appreciate your suggestion about improving the genetic model. We think the Chinese Legend can figuratively help readers, especially Chinese readers to understand this model. However, we decided to adopt the reviewer's suggestion and use a more direct and clear model. In summary, we refer to the classic genetic model of the *Ga1* locus (Lu et al., 2020; Ref. 4), and improved it accordingly in Fig. 3h-k, and hope it is more clear and precise now. We still would like to add the Chinese legend as supplementary material for more additional information.

Fig. 3 | Genetic relationship between three types of genes of the *Ga1* locus. a, Quantitative expression analysis of *ZmPRP3* and *ZmPME3* in silks of R^{SK-Z58} , R^{SK-SK} , R^{H-Z58} , and R^{H-SK} . Error bars represent mean + SD ($n = 3$), a and b indicate that means differ according to the LSD test ($P < 0.01$). **b,** Comparison of the longest pollen tubes of Z58 at 8 hours after pollination in silks of

R^{SK-Z58} and R^{SK-SK} silks, and in R^{H-Z58} and R^{H-SK}. Error bars represent mean + SD, ** $P < 0.01$, ns (not significant, $P > 0.05$, Student's t test). Numbers in each column indicate sample size. **c**, Quantitative expression analysis of *ZmPME3* in silks of WT, *ZmPME3*^{OE+}, *CR-Zmprp3*, and *ZmPME3*^{OE+}/*CR-Zmprp3*. #2 represents *ZmPME3*^{OE+} plant. H1 and H2 represent two *ZmPME3*^{OE+}/*CR-Zmprp3* plants. Error bars represent mean + SD ($n = 3$). a, b and c indicate that means differ according to the LSD test ($P < 0.01$). **d**, Comparison of the longest pollen tubes of WT [KN5585 (*ga1*)] at 8 hours after pollination in silks of *CR-Zmprp3*, *ZmPME3*^{OE+}, WT, SK and *ZmPME3*^{OE+}/*CR-Zmprp3* plants. Error bars represent mean + SD, a, b and c indicate that means differ according to the LSD test ($P < 0.01$). Numbers in each column indicate sample size. **e**, Quantitative expression analysis of *ZmPME3* in silks of *CR-Zmprp3*, *ZmPME3*^{OE+}, WT, SK and *ZmPME3*^{OE+}/*CR-Zmprp3* plants. H3-H8 represents six *ZmPME3*^{OE+}/*CR-Zmprp3* plants. #4-#7 represents four *ZmPME3*^{OE+} plant. Error bars represent mean + SD ($n = 3$), a, b and c indicate that means differ according to the LSD test ($P < 0.01$). **f**, Statistical analysis of the seed set ratio from crosses by using SK, *ZmPME3*^{OE+}, *CR-Zmprp3*, *ZmPME3*^{OE+}/*CR-Zmprp3* and WT each as a female parent, WT [KN5585 (*ga1*)] as a common male parent. H3-H8 represents six *ZmPME3*^{OE+}/*CR-Zmprp3* plants. #4-#7 represents four *ZmPME3*^{OE+} plants. Numbers in each column indicate sample size. **g**, Crossing experiments showing ears of WT after self-pollination and crosses when SK, *ZmPME3*^{OE+}, *CR-Zmprp3*, *ZmPME3*^{OE+}/*CR-Zmprp3* were used as female parents, WT [(KN5585, *ga1*)] pollen was used as male parent. Scale bar = 2 cm. **h**, *ga1* and *Ga1-M* silk-expressed *ZmPRP3* acts as an accelerator, promoting *ga1* pollen tube growth. **i**, Silk-expressed *ZmPME3* exists in *Ga1-S* silk, and is a female determinant that blocks *ga1* pollen tube growth. **j**, *ZmPMEs-m* genes are male determinants, and exist in *Ga1-S* and *Ga1-M* pollen that can completely overcome block effect of *ZmPME3*. **k**, In the heterozygous background of *ZmPME3*, introducing *ZmPRP3* can effectively break the block effect of *ZmPME3*, promote *ga1* pollen tube growth.

Q3: The GWAS results presented in this study are a minor result, but I am surprised that they only detect signals on chromosome 4 (Sup Fig 4b). Hurst et al., 2019 detected 7 traits on five chromosomes.

[Response]: Thank you for the nice comments. We have also paid attention to the mentioned article (Hurst et al., 2019). We think the reasons for the different GWAS results is the different purpose in the two studies. In our study, we used *Ga1-S* (CML304) as female parent, the inbred lines from the association mapping panel as male parents and took the CML304 seed set ratio as a phenotype for GWAS (see Materials and Methods). The experimental design was to detect the **male determinant**. Zhang et al. (2018) used the same strategy to detected signals of the male determinant and obtained similar results. Hurst et al. (2019) used a diverse panel of 311 popcorn lines as female parents and a purple *ga1* line as male parents to test whether specific loci outside the *Ga1* locus were associated with variation in the effectiveness of the

female determinant function. In summary, it is not a surprising that Hurst et al. identified different loci. Hence, we agree that the GWAS results are a minor result in present study.

Q4: The report of multiple PMEs (*ZmPmes-m*) in addition to Ga1P (or instead of?) in the pollen of Ga1s is inconsistent with the published literature (and unpublished data I have). The authors do not adequately acknowledge this discrepancy and as such, do not adequately show that these are unique functional genes rather than pseudogenes which have been previously reported. It seems more that they have identified variants of the Ga1P found in SDGa25 (and Hp301). The transgenics seem to be a good experiment, but the results are not a very strong case, especially when considering that transgenic Ga1P pollen leads to full seed set on Ga1s silks. In fact, I'd say that they almost show that the *ZmPMEs-m* variants aren't good candidates for the male factor. They only indicate one primer pair for the qPCR, so there's no independent confirmation that multiple transcripts are present (leading to unique proteins), as the primers would pick up all. Perhaps, their data indicates copy number variation in the SK line? It's one thing to show you can make a protein that has some function, it's another to show that the protein is there normally. Hp301 has an available genome sequence to make comparisons and strengthen the conclusions. Proteomic evidence would also go a long way in verifying multiple male PMEs from the Ga1 locus, as proteomic data from SDGa25 does not show additional GA1P variants in the pollen.

[Response]: We greatly appreciate your constructive suggestions to further study whether the five pollen-expressed *PME* genes are unique and functional genes rather than pseudogenes. Therefore we integrated and further re-examined our the data and found new evidence to support the view that the five *PME* genes are all expressed and functional genes on the SK genome. The details are listed as follows:

(1) We aligned 1.5K bp upstream and downstream sequences of each of the five *PME* genes and found that they exhibit high sequence identity, indicating that they contain the same regulatory cis-elements and this explains that the five *PME* genes maintain the same expression pattern (**Response Fig. 1**).

(2) We re-examined the RNA-seq data from SK pollen with three replications and found that each *PME* gene generates only one transcript (**Response Fig. 2**).

(3) By using the unique SNPs to each of the five *PME* genes that we detected in the SK genome, we confirmed that these SNPs were also present in the RNA-seq data. SNPs in the RNA-seq data clearly distinguished *ZmPMEs-m* genes from each other and revealed their respective high expression in SK pollen (**Supplementary Fig. 11**).

(4) We aligned the *Ga1* locus derived from the SK genome with the published HP301 genome (*Ga1-S* genotype; Hufford et al., 2021; **Ref. 5**), and found that there are also

five *PME* genes in the HP301 genome (**Supplementary Fig. 12**), those genes share high sequence identity to *ZmPMEs-m* with only a few SNPs differences (**Response Fig. 3**).

These results demonstrated that *ZmPMEs-m* are five expressed genes exist in *Ga1-S* genomes. In the transgenic experiment, we over-expressed each *ZmPMEs-m* genes **individually** using the ubiquitin promoter in the KN5585 (*ga1*) background, and found that pollen tubes grew faster than WT in all over-expressing events of the five *PME* genes (Fig. 2b). Furthermore, SK plants fertilized with pollen from transgenic lines over-expressing *ZmPME6* exhibited a significant increase in harvested seed number compared with WT pollen (completely sterile). We therefore concluded that each of the *ZmPMEs-m* are unique and represent functional genes that accelerate pollen tube growth. Moreover, we also noticed that transgenic *ZmGa1P* pollen lead to full seed set on *Ga1-S* silks. We think this is due to the fact that gene expression differences occur in different transgenic backgrounds and events. Zhang et al. (2018) cloned a 6,287-bp genomic fragment of Zm00001d048936 from SDGa25 and introduced it into the inbred line B104 (*ga1*). All SDGa25 plants fertilized with pollen from the transgenic plants exhibited full seed-set. The 6,287 bp sequence contains *ZmGa1P* and its native promoter, the expression level of *ZmGa1P* in B104 reached over 50% of that found in SDGa25 pollen, which exhibit higher expression level compared to the lines we generated (we detected only 8% to 13% expression level compared to that of SK pollen). Combing the above results, we speculate that the low-level expression of *PME* genes is due to the usage of the weaker ubiquitin promoter that is not capable to drive a high-level expression of *PME* gene in pollen like the native promoter (He et al., 2009; Jiang et al., 2018; **Ref. 6, 7**). We have specifically addressed the reason of low expression now in the revision of the article. We hope you are satisfied with our explanations.

Response Fig. 1 | a, Sequence of 1.5Kb upstream and downstream of the five *PME* genes. Synteny blocks are highlighted by flesh lines. **b**, Gene structure of thirteen

ZmPME genes. White boxes represent coding sequence. The *PME* domain is highlighted with a red box, and UTR regions are highlight with grey boxes. **c**, Expression level of thirteen *ZmPME* genes in nine SK tissues. FPKM, Fragments Per Kilobase per Million mapped reads.

Response Fig. 2 | RNA-seq Coverage track of the five PME genes using the Integrative Genomics Viewer (IGV). Three tracks present RNA-seq data from SK pollen with three replications.

a

```

ZnPME5 TCGGTTCCGGAGAAAGCTGTTACTCGGTAGAAGCAAGCCCTTTCATCACCATAATGTCCGAGGACCCCATGAACCCCTGCTG 400
ZnPME6 TGTGTCCGGAGAAAGCTGTTACTCGGTAGAAGCAAGCCCTTTCATCACCATAATGTCCGAGGACCCCATGAACCCCTGCTG 400
ZnPME8 TGTGTCCGGAGAAAGCTGTTACTCGGTAGAAGCAAGCCCTTTCATCACCATAATGTCCGAGGACCCCATGAACCCCTGCTG 400
ZnPME11 TGTGTCCGGAGAAAGCTGTTACTCGGTAGAAGCAAGCCCTTTCATCACCATAATGTCCGAGGACCCCATGAACCCCTGCTG 400
ZnPME13 TGTGTCCGGAGAAAGCTGTTACTCGGTAGAAGCAAGCCCTTTCATCACCATAATGTCCGAGGACCCCATGAACCCCTGCTG 400
ZnGa1P TGTGTCCGGAGAAAGCTGTTACTCGGTAGAAGCAAGCCCTTTCATCACCATAATGTCCGAGGACCCCATGAACCCCTGCTG 400
  
```

b

```

ZnPME5 GGTCCGGATGTCCAGACAGTGGAAAGGTACCCTCCTATGTCCATTACATCCTCTCCTTTCTTCATATATGATTGTGTGA 1040
ZnPME6 GGTCCGGATGTCCAGACAGTGGAAAGGTACCCTCCTATGTCCATTACATCCTCTCCTTTCTTCATATATGATTGTGTGA 1040
ZnPME8 GGTCCGGATGTCCAGACAGTGGAAAGGTACCCTCCTATGTCCATTACATCCTCTCCTTTCTTCATATATGATTGTGTGA 1040
ZnPME11 GGTCCGGATGTCCAGACAGTGGAAAGGTACCCTCCTATGTCCATTACATCCTCTCCTTTCTTCATATATGATTGTGTGA 1040
ZnPME13 GGTCCGGATGTCCAGACAGTGGAAAGGTACCCTCCTATGTCCATTACATCCTCTCCTTTCTTCATATATGATTGTGTGA 1040
ZnGa1P GGTCCGGATGTCCAGACAGTGGAAAGGTACCCTCCTATGTCCATTACATCCTCTCCTTTCTTCATATATGATTGTGTGA 1040
  
```

C

ZmPME5 TTTTATGAGGAATGCAAAATCGTTTCGGTGTTGAAGGAGGCATTGGTATTGCCATTGCCAOCACCGGAGCAGGACCGCTCTAG 803
 ZmPME6 TTTTATGAGGAATGCAAAATCGTTTCGGTGTTGAAGGAGGCATTGGTATTGCCATTGCCAOCACCGGAGCAGGACCGCTCTAG 803
 ZmPME8 TTTTATGAGGAATGCAAAATGTTTCGGTGTTGAAGGAGGCATTGGTATTGCCATTGCCAOCACCGGAGCAGGACCGCTCTAA 803
 ZmPME11 TTTTATGAGGAATGCAAAATCGTTTCGGTGTTGAAGGAGGCATTGGTATTGCCATTGCCAOCACCGGAGCAGGACCGCTCTAG 803
 ZmPME13 TTTTATGAGGAATGCAAAATGTTTCGGTGTTGAAGGAGGCATTGGTATTGCCATTGCCAOCACCGGAGCAGGACCGCTCTAG 803
 ZmGa1P TTTTATGAGGAATGCAAAATCGTTTCGGTGTTGAAGGAGGCATTGGTATTGCCATTGCCAOCACCGGAGCAGGACCGCTCTAG 803

d

ZmPME5 ACGAOCAGACGGGCTGCACTACTTCAAGGCTTGTGCCATCAAGGGAACCATCGACTTATCTTCGGATCTGCCAAGTCA 720
 ZmPME6 ACGAOCAGACGGGCTGCACTACTTCAAGGCTTGTGCCATCAAGGGAACCATCGACTTATCTTCGGATCTGCCAAGTCA 720
 ZmPME8 ACGAOCAGACGGGCTGCACTACTTCAAGGCTTGTGCCATCAAGGGAACCATCGACTTATCTTCGGATCTGCCAAGTCA 720
 ZmPME11 ACGAOCAGACGGGCTGCACTACTTCAAGGCTTGTGCCATCAAGGGAACCATCGACTTATCTTCGGATCTGCCAAGTCA 720
 ZmPME13 ACGAOCAGACGGGCTGCACTACTTCAAGGCTTGTGCCATCAAGGGAACCATCGACTTATCTTCGGATCTGCCAAGTCA 720
 ZmGa1P ACGAOCAGACGGGCTGCACTACTTCAAGGCTTGTGCCATCAAGGGAACCATCGACTTATCTTCGGATCTGCCAAGTCA 720

e

```
ZmPME5 CAAAAAGGTCTTTTTCAACTTATGGGTGACAAACCAAGCCAGCTAATGCCACCAAGATGCGGGGTGTGCTAAGAAAGATG 160
ZmPME6 CAAAAAGGTCTTTTTCAACTTATGGGTGACAAACCAAGCCAGCTAATGCCACCAAGATGCGGGGTGTGCTAAGAAAGATG 160
ZmPME8 CAAAAAGGTCTTTTTCAACTTATGGGTGACAAACCAAGCCAGCTAATGCCACCAAGATGCGGGGTGTGCTAAGAAAGATG 160
ZmPME11 CAAAAAGGTCTTTTTCAACTTATGGGTGACAAACCAAGCCAGCTAATGCCACCAAGATGCGGGGTGTGCTAAGAAAGATG 160
ZmPME13 CAAAAAGGTCTTTTTCAACTTATGGGTGACAAACCAAGCCAGCTAATGCCACCAAGATGCGGGGTGTGCTAAGAAAGATG 160
ZmGa1P CAAAAAGGTCTTTTTCAACTTATGGGTGACAAACCAAGCCAGCTAATGCCACCAAGATGCGGGGTGTGCTAAGAAAGATG 160
```

Supplementary Fig. 11 | SNPs in the RNA-seq data of SK pollen. a, Top, a unique SNP of *ZmPME5* was detected in the SK genome. Bottom, the same SNP in RNA-seq data from SK pollen (with three replications). **b,** Top, a unique SNP of *ZmPME6* was detected in the SK genome. Bottom, the same SNP in RNA-seq data from SK pollen (with three replications). **c,** Top, a unique SNP of *ZmPME8* was detected in the SK genome. Bottom, the same SNP in RNA-seq data from SK pollen (with three replications). **d,** Top, a unique SNP of *ZmPME11* was detected in the SK genome. Bottom, the same SNP in RNA-seq data from SK pollen (with three replications). **e,** Top, a unique SNP of *ZmPME13* was detected in the SK genome. Bottom, the same SNP in RNA-seq data from SK pollen (with three replications).

Supplementary Fig. 12 | Genomic structure of the *Ga1* locus compared between SK, HP301 and B73 genomes. a, Sequence alignment of the *Ga1* locus between HP301 and SK genomes. Synteny blocks are highlighted by gray lines. Annotated genes of the two genomes are highlighted by blue and red boxes. Five *PME* genes of the SK genome and five homologous genes of the HP301 genome are highlight by red boxes. **b**, Sequence alignment of the *Ga1* locus between HP301 and B73 genomes. Synteny blocks are highlighted by gray lines. Annotated genes of the two genomes are highlighted by blue and red boxes. Red boxes and “*” highlighted in red represents five *PME* genes of the SK genome and one homologous gene on the B73 genome.

Zm00027a029665	gcac t caagact t gcacaat cgaggggagaaggagaaaaaaat t t act t gggg agggg gggcacgcct gt ga	910
Zm00027a029670	gcac t caagact t gcacaat cgagggggaaggagaaaaaaat t t act t aggt agggg gggcacgcct gt ga	910
Zm00027a029667	gcac t caagact t gcacaat cgagggggaaggagaaaaaaat t t act t gggg agggg gggcacgcct gt ga	910
Zm00027a029671	gcac t caagact t gcacaat cgagggggaaggagaaaaaaat t t act t gggg agggg gggcacgcct gt ga	910
Zm00027a029679	gcac t caagact t gcacaat cgagggggaaggagaaaaaaat t t act t gggg agggg gggcacgcct gt ga	910
ZnPME5	gcac t caagact t gcacaat cgagggggaaggagaaaaaaat t t act t gggg agggg gggcacgcct gt ga	910
ZnPME8	gcac t caagact t gcacaat cgagggggaaggagaaaaaaat t t act t aggt agggg gggcacgcct gt ga	910
ZnPME6	gcac t caagact t gcacaat cgagggggaaggagaaaaaaat t t act t gggg agggg gggcacgcct gt ga	910
ZnPME11	gcac t caagact t gcacaat cgagggggaaggagaaaaaaat t t act t gggg agggg gggcacgcct gt ga	910
ZnPME13	gcac t caagact t gcacaat cgagggggaaggagaaaaaaat t t act t gggg agggg gggcacgcct gt ga	910
ZmGa1P	gcac t caagact t gcacaat cgagggggaaggagaaaaaaat t t act t gggg agggg gggcacgcct gt ga	910

Zm00027a029665	t at act cct acact aat at aggt aaggagat t gt aggcac aat at ct gat ggt cgggat gt ccagacagt	980
Zm00027a029670	t at act cct acact aat at aggt aaggagat t gt aggcac aat at ct gat ggt cgggat gt ccagacagt	980
Zm00027a029667	t at act cct acact aat at aggt aaggagat t gt aggcac aat at ct gat ggt caggacgt ccagacagt	980
Zm00027a029671	t at act cct acact aat at aggt aaggagat t gt aggcac aat at ct gat ggt cgggacgt ccagacagt	980
Zm00027a029679	t at act cct acact aat at aggt aaggagat t gt aggcac aat at ct gat ggt cgggacgt ccaaacagt	980
ZnPME5	t at act cct acact aat at aggt aaggagat t gt aggcac aat at ct gat ggt cgggat gt ccagacagt	980
ZnPME8	t at act cct acact aat at aggt aaggagat t gt aggcac aat at ct gat ggt cgggat gt ccagacagt	980
ZnPME6	t at act cct acact aat at aggt aaggagat t gt aggcac aat at ct gat ggt caggacgt ccagacagt	980
ZnPME11	t at act cct acact aat at aggt aaggagat t gt aggcac aat at ct gat ggt cgggacgt ccagacagt	980
ZnPME13	t at act cct acact aat at aggt aaggagat t gt aggcac aat at ct gat ggt cgggacgt ccaaacagt	980
ZmGa1P	t at act cct acact aat at aggt aaggagat t gt aggcac aat at ct gat ggt cgggacgt ccagacagt	980

Zm00027a029665	cgaaagg accact cct at gt ccat t acat cct ct cct t t ct t cat at at gat t gt gt gat t aaaggt gt	1050
Zm00027a029670	cgaaagg accact cct at gt ccat t acat cct ct cct t t ct t cat at at gat t gt gt gat t aaaggt gt	1050
Zm00027a029667	cgaaagg accact cct at gt ccat t acat cct ct cct t t ct t cat at at gat t gt gt gat t aaaggt gt	1050
Zm00027a029671	cgaaagg accact cct at gt ccat t acat cct ct cct t t ct t cat at at gat t gt gt gat t aaaggt gt	1050
Zm00027a029679	cgaaagg accact cct at gt ccat t acat cct ct cct t t ct t cat at at gat t gt gt gat t aaaggt gt	1050
ZnPME5	cgaaagg accact cct at gt ccat t acat cct ct cct t t ct t cat at at gat t gt gt gat t aaaggt gt	1050
ZnPME8	cgaaagg accact cct at gt ccat t acat cct ct cct t t ct t cat at at gat t gt gt gat t aaaggt gt	1050
ZnPME6	cgaaagg accact cct at gt ccat t acat cct ct cct t t ct t cat at at gat t gt gt gat t aaaggt gt	1050
ZnPME11	cgaaagg accact cct at gt ccat t acat cct ct cct t t ct t cat at at gat t gt gt gat t aaaggt gt	1050
ZnPME13	cgaaagg accact cct at gt ccat t acat cct ct cct t t ct t cat at at gat t gt gt gat t aaaggt gt	1050
ZmGa1P	cgaaagg accact cct at gt ccat t acat cct ct cct t t ct t cat at at gat t gt gt gat t aaaggt gt	1050

Zm00027a029665	t gt t cat t at ct at act gat gaaaggt gt t gt t gt t gcgt t gcat t t t t at t at agggggt act act gc	1120
Zm00027a029670	t gt t cat t at ct at act gat gaaaggt gt t gt t gt t gcgt t gcat t t t t at t at agggggt act act gc	1120
Zm00027a029667	t gt t cat t at ct at act gat gaaaggt gt t gt t gt t gcgt t gcat t t t t at t at agggggt act act gc	1120
Zm00027a029671	t gt t cat t at ct at act gat gaaaggt gt t gt t gt t gcgt t gcat t t t t at t at agggggt act act gc	1120
Zm00027a029679	t gt t cat t at ct at act gat gaaaggt gt t gt t gt t gcgt t gcat t t t t at t at agggggt act act gc	1120
ZnPME5	t gt t cat t at ct at act gat gaaaggt gt t gt t gt t gcgt t gcat t t t t at t at agggggt act act gc	1120
ZnPME8	t gt t cat t at ct at act gat gaaaggt gt t gt t gt t gcgt t gcat t t t t at t at agggggt act act gc	1120
ZnPME6	t gt t cat t at ct at act gat gaaaggt gt t gt t gt t gcgt t gcat t t t t at t at agggggt act act gc	1120
ZnPME11	t gt t cat t at ct at act gat gaaaggt gt t gt t gt t gcgt t gcat t t t t at t at agggggt act act gc	1120
ZnPME13	t gt t cat t at ct at act gat gaaaggt gt t gt t gt t gcgt t gcat t t t t at t at agggggt act act gc	1120
ZmGa1P	t gt t cat t at ct at act gat gaaaggt gt t gt t gt t gcgt t gcat t t t t at t at agggggt act act gc	1120

Zm00027a029665	gccact t t caggt gt t acgggcct gggat gt ct ccaat ggt aacct caact ct gacct at gt cgaggcaa	1190
Zm00027a029670	gccact t t caggt gt t acgggcct gggat gt ct ccaat ggt aacct caact ct gacct at gt cgaggcaa	1190
Zm00027a029667	gccact t t caggt gt t acgggcct gggat gt ct ccaat ggt aacct caact ct gacct at gt cgaggcaa	1190
Zm00027a029671	gccact t t caggt gt t acgggcct gggat gt ct ccaat ggt aacct caact ct gacct at gt cgaggcaa	1190
Zm00027a029679	gccact t t caggt gt t acgggcct gggat gt ct ccaat ggt aacct caact ct gacct at gt cgaggcaa	1190
ZnPME5	gccact t t caggt gt t acgggcct gggat gt ct ccaat ggt aacct caact ct gacct at gt cgaggcaa	1190
ZnPME8	gccact t t caggt gt t acgggcct gggat gt ct ccaat ggt aacct caact ct gacct at gt cgaggcaa	1190
ZnPME6	gccact t t caggt gt t acgggcct gggat gt ct ccaat ggt aacct caact ct gacct at gt cgaggcaa	1190
ZnPME11	gccact t t caggt gt t acgggcct gggat gt ct ccaat ggt aacct caact ct gacct at gt cgaggcaa	1190
ZnPME13	gccact t t caggt gt t acgggcct gggat gt ct ccaat ggt aacct caact ct gacct at gt cgaggcaa	1190
ZmGa1P	gccact t t caggt gt t acgggcct gggat gt ct ccaat ggt aacct caact ct gacct at gt cgaggcaa	1190

Zm00027a029665	t accct t t ct cgggat acact acat ct cgggggagt cat ggat cccgt cctt accaccgct gaagaat aa	1261
Zm00027a029670	t accct t t ct cgggat acact acat ct cgggggagt cat ggat cccgt cctt accaccgct gaagaat aa	1261
Zm00027a029667	t accct t t ct cgggat acact acat ct cgggggagt cat ggat cccgt cctt accaccgct gaagaat aa	1261
Zm00027a029671	t accct t t ct cgggat acact acat ct cgggggagt cat ggat cccgt cctt accaccgct gaagaat aa	1261
Zm00027a029679	t accct t t ct cgggat acact acat ct cgggggagt cat ggat cccgt cctt accaccgct gaagaat aa	1261
ZnPME5	t accct t t ct cgggat acact acat ct cgggggagt cat ggat cccgt cctt accaccgct gaagaat aa	1261
ZnPME8	t accct t t ct cgggat acact acat ct cgggggagt cat ggat cccgt cctt accaccgct gaagaat aa	1261
ZnPME6	t accct t t ct cgggat acact acat ct cgggggagt cat ggat cccgt cctt accaccgct gaagaat aa	1261
ZnPME11	t accct t t ct cgggat acact acat ct cgggggagt cat ggat cccgt cctt accaccgct gaagaat aa	1261
ZnPME13	t accct t t ct cgggat acact acat ct cgggggagt cat ggat cccgt cctt accaccgct gaagaat aa	1261
ZmGa1P	t accct t t ct cgggat acact acat ct cgggggagt cat ggat cccgt cctt accaccgct gaagaat aa	1261

Response Fig. 3 | Genomic sequence alignment of *ZmPMEs-m* genes, five annotated *PME* genes on the HP301 genome and of *ZmGa1P*. “*” indicates identical nucleotides. The two exons in all *PME* genes are highlighted with red boxes. The sequence of *PME* genes that are derived from the HP301 genome are highlighted by gray boxes.

Q5: There are several instances of grammatical errors, but not worth mentioning as a

major rewrite is in order. The references 1-45 are separated from the remainder 46-61 by methods, acknowledgments, etc. Line 119: this is the first report that I can find of SK and CML304 being Ga1s. If not, please reference.

[Response]: We greatly appreciate your suggestions. We have revised the text carefully, and tried our best to correct grammatical error including a thorough review by one of our co-authors who is a native speaker. We hope that you will be satisfied with our modifications. If the reviewer think we have grammatical errors in the text that affect understanding, please help us to pinpoint them, we would appreciate it. Additionally, reference 1-51 were cited in Introduction, Result and Discussion, while, reference 52-67 were cited in Method, we have now combined all references at the end of the revised manuscript. As far as we know, SK and CML304 have never been reported before as a *Ga1-S* genotype.

Q6: Validity: The methods are sound, although the interpretations are sometimes questionable or identical to previous work and not properly credited.

[Response]: We apologize for insufficient citations. we have now added eight related papers. Details of citations are listed in **Response Q8**. If the reviewer still thinks we have inappropriate citations or unclear interpretations that affect understanding, please help us to pinpoint them,we would appreciate it.

Q7: Most of the work can be reproduced, however, the genome sequence for SK cannot be accessed in the U.S. At least the link does not work from their Yang 2019 publication.

[Response]: We apologize for this. The SK genome can now be obtained from <https://download.maizegdb.org/Zm-SK-REFERENCE-YAN-1.0/>. This reference is now also provided in the manuscript.

Q8: Specific instances where results are the same as published reports, but the authors do not make clear that their results are confirming published observations.

Pollen tube measurement conclusions were the same as discussed in Zhang, et al 2012 (their reference 8). The authors chose later time points but came to the same conclusion **without acknowledging that**. This was also reported (and adequately acknowledged) in **Moran Lauter, et al. 2017**.

(1) Pollen tube measurement conclusions were the same as discussed in Zhang, et al 2012 (their reference 8). The authors chose later time points but came to the same

conclusion **without acknowledging that**. This was also reported (and adequately acknowledged) in **Moran Lauter, et al. 2017**.

- (2) Line 102: Segregation distortion at this locus has been observed since the 1920s and reported many times. The authors' contribution is separating out component2, and this needs to be made clear.
- (3) Fig 1. ZmPme3 gene structure in Ga1s and B73 (28. Moran Lauter, et al. 2017); ZmPMEs-m shown in (14) Zhang, et al. 2018 as pseudogenes. Insertion in ZmPme3 in the Ga1m line NC390xNC394 reported in 28. Moran Lauter, et al. 2017
- (4) Line 155: Fine-mapping has been successful yielding the same interval shown here, and it should be cited (Bloom and Holland, 2011; 8. Zhang et al., 2012, 21. Liu et al., 2014) Some of the primers used in this study are identical to those used in the Liu 2014 study.
- (5) Line 179: it isn't made clear that Zm00001d048936 is the non-functional B73 version of Ga1P that is reported in 14. Zhang et al, 2018. Line 191: 14. Zhang et al., 2018 showed a different variation between B73 and Ga1P. This should be noted.
- (6) Lines 210-218: all reported in 28. Moran Lauter, et al. 2017 and not cited.
- (7) qPCR and Resequencing primers are from 28. Moran Lauter, et al. 2017 (therefore, they had to know that they were repeating this work)
- (8) Supplemental Fig 12 is reported in 29. Lu, et al. 2019 (Sup Fig 7)
- (9) Model in 3h is a variation on Lu, et al. Plant Reproduction 33, pages 117–128 (2020) (paper not cited in the manuscript)

My summary: The main discovery in this work is a new and important role for ZmPRP3 in the regulation of pollination, and its ability to block ZmPme3 action in Ga1s gametophytic incompatibility. This discovery allows the authors to confirm that ZmPme3 is the causal silk factor in Ga1s UCI. Their data suggests that some previously reported Ga1P copies are actually transcribed but don't quite show a functional role. There are several instances of data confirming published results, which do not add knowledge to the field.

[Response]: Thank you very much for the nice summary. We apologize for the insufficient citation. We have carefully reviewed the research paper related to maize cross incompatibility for nearly two decades, to ensure that all related publications are correctly and appropriately cited. We have added eight related papers as the reviewer suggested, details of citations are listed as follows. We thank all the scientists who have contribute to maize cross incompatibility field, and firmly believe that their work should be appreciated and respected.

- (1) Pollen tube measurement conclusions are cited from Zhang et al. 2012 (**Ref. 8**) and

Moran Lauter et al. 2017 (Ref. 9). In addition, we added the results of *in vivo* pollen tube growth visualized by aniline blue staining and found that Zheng58 pollen tube arrest growing before reach the ovule, which is consistent with Lu et al. 2013 (Ref. 4).

- (2) We have improved the description of fine mapping work, and pointed out our contribution in separating component2 from the *Ga1* fine mapping interval.
- (3) *ZmPME3* gene structure and insertion in the *Ga1-M* of NC390xNC394 is now cited from Moran Lauter et al. 2017 (Ref. 9). We have added the citation to the relevant results and also added new result to prove *ZmPMEs-m* are five unique, functional genes in the revised manuscript (see detail in **Response Q4**).
- (4) Fine mapping interval of the *Ga1* locus is cited from Bloom and Holland, 2011 (Ref. 10), Zhang et al. 2012 (Ref. 8) and Liu et al. 2014 (Ref. 11). Molecular markers M1, M5 and M6 that we used for fine mapping the *Ga1* locus are cited from Bloom and Holland, 2011 (Ref. 10), M5 and M6 are cited using reference Liu et al. 2014 (Ref. 11). We have also added the citations to the Methods.
- (5) We have adjusted the description of *ZmGa1P* in the revised manuscript, to made clear that Zm00001d048936 is the non-functional B73 version of *ZmGa1P* and pointed out the different variation between B73 and *ZmGa1P* that is reported in Zhang et al. 2018 (Ref. 12).
- (6) Re-sequencing and qPCR results of *ZmPME3* in Line210-218 are consistent with Moran Lauter et al. 2017 (Ref. 9). We have added the citations to the relevant results in the revised manuscript.
- (7) Primers used for re-sequencing *ZmPME3* are cited from Moran Laute, et al. 2017 (Ref. 9), and we have added the citations to the Methods.
- (8) The conclusion that *ZmPME3* shares high similarity with *Tcb1-f* is cited from Lu et al. 2019 (Ref. 13), and we have added the citation to the Introduction.
- (9) On the basis of the classic genetic model of the *Ga1* locus cited from Lu et al. 2020 (Ref. 14), we have pointed out the newly found silk-expressed *ZmPRP3* that exists in *ga1* and *Ga1-M* silks and acts as an accelerator of pollen tube growth. A modified model was added now as described in **Response Q2**.

Response to Reviewer#2:

Wang et al., perform a genetic analysis to uncover the genetic determinants regulating unilateral cross compatibility in maize in the male and female. As a result of their

extensive genetic analysis the authors identify a group of pollen expressed PME3 and a silk-expressed PME3 and cysteine-rich protein. To validate the role of these genes in the control of UCI, the authors use CRISPR-cas9 and overexpression approaches to create different gene combinations with the aim to determine the function of the different candidate genes in the control of pollen tube growth. In addition, the authors also investigated the methylesterification status of cell walls of silk cells in different genetic backgrounds. Wang et al., in their study, they also investigate the evolutionary origin of the Gal1 locus.

The authors have done an extensive genetic analysis and used proper genetic validation tools to characterize their candidate genes. In their model the authors suggest that activation of PME3 may lead to the condensation of silk cell walls creating a physical barrier

Thank you for your comprehensive summary and praise. We have also seriously considered your criticism and hope that the revised version can satisfy you.

Q1: a) The role of PMEs in the regulation of pollen dynamic and growth has been already described in other model species like Arabidopsis. I think the Introduction will benefit from including that with a few relevant references (as an example : Bosh et al., Plant Cell, 2005).

[Response]: We greatly appreciate this suggestion, we have included the citations now for the functional description of the *PME* gene.

Q2: b) The authors check the esterification status if silk cells using LM20, and mainly binds to methyl esterified pectins. In their model they suggest that PME3 activation may lead to condensation of the cell wall establishing a physical barrier for growing pollen tubes. Have the authors checked with the 2F4 antibody (binds to Ca egg-box pectin) whether in the case of their PME3Ox lines the pattern is different compared to ZmPRP3, Wt and SK genotypes?

[Response]: We greatly appreciate this constructive and helpful suggestions on the core topic about checking the esterification status of cells in *ZmPME3^{OE+}*, *ZmPRP3*, WT and SK silks. This is very important to further investigate the molecular function of *ZmPME3* and *ZmPRP3*. We agree that it is necessary to confirm the esterification status of silk cells with an antibody other than LM20 and appreciate the suggestion of using the 2F4 antibody. However, we regret to be informed that the 2F4 antibody is discontinued for sale in China and we cannot obtain it. Considering that our purpose is to check the esterification status of silk cells, we thought it should be acceptable to choose the LM19 antibody as an alternative to 2F4, it functions similarly to 2F4, and can be used to labeled low-methylesterified homogalacturonans (HGs; Xu et al., 2011; Ref. 15). We confirmed that WT and CR-*Zmprp3* silk cells exhibited stronger

fluorescence intensity compared to SK and *ZmPME3^{OE+}* (**Supplementary Fig. 25**), which is contrary to the LM20 result as expected (**Supplementary Fig. 24**). This proved that the esterification status of silk cells of SK and *ZmPME3^{OE+}* are different from that of WT and *CR-Zmprp3*, showing lower degree of methyl esterified pectin. A minimum stretch of nine de-esterified GalA residues can form Ca²⁺ linkages, which may eventually strengthen the cell wall (Wolf et al., 2009; **Ref. 16**). We have added these new results in the revised manuscript, to support our hypothesis that *ZmPME3* activity solidifies the silk cell walls as a major mechanism to establishes a barrier for growing pollen tubes.

Supplementary Fig. 25 | Immuno-detection of methylesterified pectin in silk cells. a, Immuno-detection of methylesterified pectin in silk cells of SK (*Ga1-S*), WT [KN5585 (*ga1*)], *ZmPME3^{OE+}* and *CR-Zmprp3* using the LM19 antibody. Scale bar = 50 μ m, FITC, Fluorescein isothiocyanate. **b,** Quantification of fluorescent signal intensity in silk cells shown in (**a**). Error bars represent mean + SD. a, b indicate that means differ according to the LSD test ($P < 0.01$). Numbers in each column indicate sample size.

Q3: c) When the authors described that pollen tube stop growing. Do they pollen tubes burst? or they just maintain their integrity but stop growing?

[Response]: We apologize that this has not being adequately described and explained in the previous manuscript. In fact, we observed Z58 pollen tube reached only 30% of the total length of SK silks at 24 hours after pollination, with a heavy callose deposition at the tip and arrest growing before reach the ovule. We have added the results of *in vivo* pollen tube growth visualized by aniline blue staining into **Supplementary Fig. 1c**. Since pollen tubes growth arrest occurs very far from the ovule, we assume that pollen tubes maintain their integrity but stop growing.

Supplementary Fig. 1 | Seed set of crosses between SK and Z58. (a) Comparison of the longest pollen tube at 8 hours after pollination in silks of SK, using pollen of SK and Z58, respectively. Error bars represent mean + SD, **P < 0.01 (Student's t test). Numbers in each column indicate the sample size. **(b)** Comparison of the longest pollen tube at 24 hours after pollination in silks of SK, using pollen of SK and Z58, respectively. Error bars represent mean + SD, **P < 0.01 (Student's t test). Numbers in each column indicate the sample size. **(c) Morphology of SK and Z58 pollen tubes in SK silks. Arrows point to normal SK pollen tubes, arrowhead indicates a heavy callose deposition at the tip of Z58 pollen tubes. Scale bar = 100µm. (d)** Longest pollen tube as a percentage of the total silk length. Error bars represent mean + SD, **P < 0.01 (Student's t test). Numbers in each column indicate sample size. **(e)** Crossing experiments showing ears of self-pollinated SK (Ga1-S) and crosses between the SK and Z58 (ga1) after SK silks were cut shorter than 5cm. Scale bar = 2 cm

Q4: d) It would be good that the authors would include the independent data points in their bar graphs, together with the error.

[Response]: We greatly appreciate this suggestions, we have now added the independent data points in every bar graphs to present our data more clearly.

Q5: e) the authors should also check what type of data population they are analyzing.

Most likely a t-test would not be the best tool. If that would be the case it should be described and justified in the methods.

[Response]: We greatly appreciate for reminding us to recheck the statistical methods used in this study.

Throughout the study, **we used the Student's t test in the following cases**

- (1) Comparing the pollen tube growth rate in silks of WT and *CR-Zmprp3* silks.
- (2) Comparing the seeding ratio of WT and *CR-Zmprp3* when used WT as male parent respectively.
- (3) Comparing the growth rate of SK and Zheng58[Z58(*ga1*)] pollen tube in SK silks at 8 hours after pollination.
- (4) Comparing the pollen tube growth rate in silks of WT[KN5585(*ga1*)] and *CR-Zm00001d048948* silks.
- (5) Comparing the pollen tube growth rate in silks of WT[KN5585(*ga1*)] and *CR-Zm00001d048949/50* silks.
- (6) Comparing the mapped reads count of *ZmPMEs-m* between *lan1* and *lan2* populations.
- (7) Comparing the mapped reads count of *ZmPMEs-m* between *par1* and *par2* populations.

Except the t-test, we also used the LSD test in the following cases:

- (1) Comparing the expression level of the *ZmPMEs-m* in WT[KN5585(*ga1*)], SK and *ZmPMEs-m*^{OE+} transgenic plant pollen; Comparing the growth rate of *ZmPMEs-m*^{OE+}, WT and SK pollen in SK silks; Comparing the seeding ratio of SK when used *ZmPME6*^{OE+}, WT and SK as male parent respectively.
- (2) Comparing the expression level of *ZmPME3* in WT, SK and transgenic plant silks; Comparing the WT[KN5585(*ga1*)] pollen tube growth rate in silks of WT, SK and *ZmPME3*^{OE+} transgenic plants; Comparing the seeding ratio of *ZmPME3*^{OE+}, WT and SK when used WT as male parent respectively.
- (3) Comparing the expression level of *ZmPME3* and *ZmPRP3* in silks of R^{SK-Z58}, R^{SK-SK}, R^{H-Z58}, and R^{H-SK} lines.
- (4) Comparing the expression level of *ZmPME3* in WT, SK, *ZmPME3*^{OE+} and *ZmPME3*^{OE+}/*CR-Zmprp3* transgenic plant silks; Comparing the WT[KN5585(*ga1*)] pollen tube growth rate in silks of WT, SK, *ZmPME3*^{OE+} and *ZmPME3*^{OE+}/*CR-Zmprp3*

transgenic plants; Comparing the seeding ratio of *ZmPME3^{OE+}*, *ZmPME3^{OE+}/CR-Zmprp3*, WT and SK when used WT as male parent respectively.

(5) Comparing the fluorescence intensity of pectin that labeled by LM20 and LM19 in WT, SK, *ZmPME3^{OE+}* and CR-*Zmprp3* silks.

When comparing the differences between two sets of data, we choose the t-test, when comparing the difference between multiple groups of data, we choose the LSD test. We consider the choice of these two methods should be appropriate in our study and are described in the figure legends, based on your suggestion, we now added more detail in the revised Methods. If the reviewer believe that we have used an inappropriate test or incorrect description, please help us to pinpoint these occurrences in order that we can correct them.

Q6: There are some grammar errors in the text. It would be good to revise the text gain. Here are two examples :

Line 100 : An segregation distortion locus that the genotypes of two parents significantly deviation from the expected ratio (1:1) was detected on chromosome 4 from 2Mb to 25Mb (Supplementary Fig. 2), which overlapped with the defined the Ga1 locus in previous reports

Line 518: ZmPMEs-m inactive (inactivates?) ZmPME3, thereby overcoming the incompatibility effect of ZmPME3.

[Response]: We greatly appreciate your suggestions, we have carefully revised the whole text, including the two parts pointed by the reviewer. We tried our best to correct the grammatical error. We hope you are satisfied with our modifications.

Response to Reviewer#3:

The main goal of the manuscript by Wang and collaborators is to identify the genetic determinants of the unilateral cross incompatibility (UCI) in the genus *Zea*. UCI has been observed both, in crosses between maize varieties and, in crosses between maize and their closest wild relatives, the teosintes. UCI acts as prezygotic reproductive isolation (RI). Dissecting mechanisms behind RI is interesting in several respects: it can illuminate the early set of reproductive barriers that may be selected to maintain the integrity of locally-adapted plants in face of maladaptive gene flow, it is also useful for the use of genetic resources in plant breeding which requires to overcome such barriers. I enjoyed reading this paper, it is well written and certainly is a great advance to our understanding of the complex genetic system underlying UCI.

The authors fine-mapped the Ga1 locus using RIL populations; they recovered complete alignments of the region from the maize reference genome B73 (ga1 type that can accept any type of pollen) and SK (Ga1-S type that cannot be pollinated by ga1 pollen) revealing extensive structural variation including 13 consecutive PME genes in SK versus a single one in B73. The authors confirmed the role of genes encoding pectin methylesterase which act as male determinants to restore pollen compatibility. They further showed that the Ga1 locus is composed of 7 expressed linked genes: five ZmPMEs-m genes expressed in the pollen, one silk-expressed ZmPME3 gene, and the newly-discovered ZmAPG1 gene. They used measures of seed set ratios from crosses involving the three haplotypes at the Ga1 locus (represented in an association mapping panel: ga1, Ga1-S, Ga1-M) and ga1 as male parent, RNA-seq, expression assays and transgenics to show that (1) Cumulative expression of ZmPMEs-m genes accelerate pollen tube growth in Ga1-S and GA1-M pollen; (2) ZmPME3 is expressed in silks of all Ga1-S lines where it blocks ga1 pollen tube growth but is either not expressed or interrupted in ga1 or Ga1-M lines; ZmAPG1 gene is highly expressed in ga1 and Ga1-M silks but absent from Ga1-S lines, and likely accelerates pollen growth in the former, although it is not sufficient alone. ZmPMEs-m genes are capable to endow Ga1-S and Ga1-M pollen become compatibility in any type of silks. Hence, pollen tube growth is determined by a balance of pectin esterification-de-esterification at the apical region of the pollen tube controlled by PME genes, while ZmAPG1 plays an auxiliary role via an independent pathway. Finally, the authors have investigated the diversity around the Ga1 locus using 771 diverse germplasm including maize and parviglumis, and further sequenced the focal genes. They found that ZmAPG1 was absent from Ga1-S lines, and that ZmPMEs-m genes were absent from ga1 lines. All the analyses are well conducted and they support the conclusions.

Thank you for your comprehensive summary and praise.

Q1: (1) The evolutionary analysis would benefit from discussion extension about what we know about cross-incompatibilities between maize and teosintes. To my knowledge, such incompatibilities are particularly strong between maize and the subspecies mexicana. I would include mexicana in the sampling.

[Response]: We greatly appreciate your constructive suggestions on evolutionary analysis, we have added 49 *Zea mays ssp. mexicana* (hereafter *mexicana*) into our evolutionary route. PC1 clearly separated *parviglumis* from maize (*landrace*, *ga1*, *Ga1-M*, *Ga1-S* type lines), PC2 separated *parviglumis* into two sub-clusters [*par1* (n = 19) and *par2* (n = 49)], *mexicana* (n=49) exhibit closer relationship to *par2* (**Response Fig4. a**). We also genotyped *ZmPMEs-m* and *ZmPRP3* genes across all 49 *mexicana* lines by calculating mapped reads count, and found that *ZmPMEs-m* and *ZmPRP3* were present in 49 *mexicana* (**Response Fig4. c**). Moreover, haplotype analysis on *ZmPME3* further showed that *par2*, *mexicana* and *lan2* possess the similar *ZmPME3* haplotype that is

most closely related to *Ga1-S* and *Ga1-M* type lines (**Response Fig4. b**). All these results suggested that *mexicana* possess only one genotype at the *Ga1* locus and is very similar to *par2*. These new results have been added to the revised manuscript.

Response Fig. 4 | Evolution and modification of the *Ga1* locus during maize domestication. a, Principal component analysis (PCA) of the *Ga1* locus in *parviglumis*, *mexicana* (*mex*), landraces, *Ga1-M*, *Ga1-S* and *ga1* inbred lines. **b**, Constructing the local haplotype clusters of *ZmPME3* in *Ga1-S*, *Ga1-M*, *lan2*, *par2* and *mexicana* (*mex*) using SNPs from 1kb sequence upstream and downstream of *ZmPME3*. **c**, Mapped reads count of *ZmPMEs-m* genes and *ZmPRP3* among two *parviglumis* sub-clusters (*par1*, *par2*), two landrace sub-clusters (*lan1*, *lan2*), *mexicana* (*mex*), *Ga1-S*, *Ga1-M* and *ga1* type lines. **P < 0.01 (Student's t test)

Q2: (2) Overall, the paper is quite complex and would benefit greatly from a figure that would summarize the results. In the current version, there is an attempt to do so in Figure 3h, but this figure could be improved and extended to present a broader overview of all the findings.

[Response]: We appreciate your suggestion about improving the genetic model. We think the Chinese Legend can figuratively help readers, especially Chinese readers to understand this model. However, we decided to adopt the reviewer's suggestion and use a more direct and clear model. In summary, we refer to the classic genetic model of the *Ga1* locus (Lu et al., 2020) (Ref. 14), and improved it in Fig. 3h-k, we hope it is now more clear and precise. We still would like to add the Chinese legend as supplementary material for more information.

Q3: (3) A careful check of all figures would be good. For instance, in figure 1b, the authors should clarify what does the table stands for (segregation distortions), that H means Heterozygous etc.. The colours in Figure 1b stands for the genomic backgrounds, but in 1C the same colours are employed to designate gene-types, this can easily get confusing for the reader.

[Response]: We appreciate this comment, and apologize for the inconvenience of reading due to inappropriate use of colors and insufficient description of legends. We have now adjusted the colors used in all figures to avoid misunderstandings.

Reference

1. Hawkins, L. K. et al. Survey of Candidate Genes for Maize Resistance to Infection by *Aspergillus flavus* and/or Aflatoxin Contamination. *Toxins (Basel)* **10**, 61 (2018).
2. Mishina, T. E. & Zeier, J. Pathogen-associated molecular pattern recognition rather than development of tissue necrosis contributes to bacterial induction of systemic acquired resistance in *Arabidopsis*. *Plant J.* **50**, 500-513 (2007).
3. Bravo, J. M. et al. Fungus- and wound-induced accumulation of mRNA containing a class II chitinase of the pathogenesis-related protein 4 (PR-4) family of maize. *Plant Mol. Biol.* **52** 745-759 (2003).
4. Lu, Y., Kermicle, J. L. & Evans, M. M. Genetic and cellular analysis of cross-incompatibility in *Zea mays*. *Plant Reproduction* **27**, 19-29 (2013).
5. Hufford, M. B. et al. De novo assembly, annotation, and comparative analysis of

- 26 diverse maize genomes. *Science* **373**, 655-662 (2021).
6. He, C. et al. Identification of a rice actin2 gene regulatory region for high-level expression of transgenes in monocots. *Plant Biotechnol.* **7**, 227-239 (2009).
 7. Jiang, P. et al. Characterization of a strong and constitutive promoter from the Arabidopsis serine carboxypeptidase-like gene *AtSCPL30* as a potential tool for crop transgenic breeding. *BMC Biotechnology* **18**, 59 (2018).
 8. Zhang, H. et al. Genetic analysis and fine mapping of the *Ga1-S* gene region conferring cross-incompatibility in maize. *Theor. Appl. Genet.* **124**, 459-465 (2012).
 9. Moran Lauter, A. N. et al. A pectin methylesterase *ZmPme3* is expressed in *Gametophyte factor1-s* (*Ga1-s*) silks and maps to that locus in maize (*Zea mays* L.). *Front Plant Sci.* **8**, 1926 (2017).
 10. Bloom, J. C. & Holland, J. M. Genomic localization of the maize cross-incompatibility gene, *Gametophyte factor 1* (*ga1*). *Maydica* **56**, 379 – 387 (2011).
 11. Liu, X. et al. Fine mapping of the maize cross-incompatibility locus *gametophytic factor 1* (*ga1*) using a homogeneous population. *Crop Sci.* **54**, 873-881 (2014).
 12. Zhang, Z. et al. A PECTIN METHYLESTERASE gene at the maize *Ga1* locus confers male function in unilateral cross-incompatibility. *Nat. Commun.* **9**, 3678 (2018).
 13. Lu, Y. X. et al. A pistil-expressed pectin methylesterase confers cross-incompatibility between strains of *Zea mays*. *Nat. Commun.* **10**, 2304 (2019).
 14. Lu, Y. X. et al. Insights into the molecular control of cross-incompatibility in *Zea mays*. *Plant Reprod.* **33**, 117-128 (2020).
 15. Xu, C. et al. Developmental localization and methylesterification of pectin epitopes during somatic embryogenesis of banana (*Musa spp.* AAA). *PLoS One* **6**, e22992 (2011).
 16. Wolf, S., Mouille, G. & Pelloux, J. Homogalacturonan methyl-esterification and plant development. *Molecular Plant* **2**, 851-860 (2009).

Reviewers' Comments:

Reviewer #1:

Remarks to the Author:

The authors have addressed most of my concerns to my satisfaction. There are a few remaining concerns, which I hope can be addressed before publication.

a) Naming Ga1P variants. There isn't an established nomenclature for CNV, but ZmGa1P is a gene with a name. The names should make it clear that these are Ga1P variants, not new pectin methylesterase genes. Giving the five ZmPmes-m genes a numerical designation is too similar to conventional PME gene names from other organisms. For example, ZmPme3 is a PME38 homolog and ZmGa1P is a PME63 homolog. Maybe something like Ga1Pa, β (just a suggestion. As I said there is currently no convention for CNV).

I would like to see discussion as to why Zhang 2018 detected one protein in SDG25a but Hp301 and SK seem to have 5 genes/copies. Is this a technical or biological difference? Does SDG25a have these 5 and the Zhang methods didn't catch them? Or is there a second allele for Ga1-S?

b) Supplementary Figure 16 (and 13): CML121 is known to be Ga1-M and CIMBL37 is known to be ga1. Are they mislabeled in your figure or do you have new data to suggest they are Ga1-S and Ga1-M, respectively?

c) Line 240-241: need to add in a caveat that not ALL Ga1-M lines do have ZmPRP3.

d) Supp Figure 5: PME10 what is the relationship to ZmPME10-1 reported in Zhang 2018? Is this a different gene or is it mislabeled?

e) Lines 159-161: However, in the corresponding Component1 region of B73, there is only one gene (Zm00001d048936), encoding a PME and exhibiting synteny (Fig. 1c, d).

This isn't wholly accurate. B73 has PME pseudogenes of both Ga1P and ZmPme3 (Zhang 2018 and Moran Lauter 2017).

Grammatical corrections:

Line 233: which encodes a small protein containing a barwin

Line 410: pollen with

Line 412: tube growth through silks

Lines 419-20: The presence of Zmprp3 may explain why a few Ga1-M lines with intact ZmPme3 genes and ZmPme3^{OE+} can still be fertilized by ga1 pollen...

Line 505: types of maize were

Line 550: Ga1-S pollen grows through the Ga1-S silk

Line 557: it is possible

Reviewer #2:

Remarks to the Author:

The authors have responded and addressed the questions raised by the reviewers.

I would say there is still something the authors should double check and it is the use of the t-test. I think that will very much depend on the type of data population they have, which they haven't described in their use of this method.

Reviewer #3:

Remarks to the Author:

I reviewed a modified version of the paper by Wang and collaborators on the genetic determinants of

the unilateral cross incompatibility (UCI) in the genus *Zea*. The authors have deeply modified their manuscript and answered my comments. I would like to acknowledge their efforts in addressing the reviewers' comments. I have however two minor comments remaining:

- Regarding the Figure 1b, I still think that the numbers presented in the Table are not clearly explained in the legend. The authors should mention from which cross these progenies are coming from, what the counts represent.

- As for Figure 4 which is a new figure that answers my first comments: I am asking them to modify the Figure 4c by discarding the use of arrows. The authors have shown that *parviglumis* encompasses two different versions of haplotypes, but they can't infer evolutionary paths. For example, Ga1-M (+-+) could have evolved from (+++) or (--+), and I see no reasons to think that the former is more likely. So, I would stay careful about the interpretations, and simply present Teosintes as encompassing two haplotypes, Landraces as two, and Modern maize as three without trying to infer an evolutive scenario. For example, it is not clear to me why the authors define the (+++) as ancestral in Figure 4d. To define an ancestral state one would need an outgroup species. Along this line, it would be good to discuss what are par1 and par2 (these clusters have likely been already described in the literature so it would help if the authors can relate them to what was discovered before, which of par1 or par2 is the cluster that corresponds to the putative domestication location?). Finally, I found the Figure 4d quite confusing. I suggest that the authors use Figure 4c instead to report the frequency of each haplotype.

Point-by-point Response to Reviewer's comments:

Response to Reviewer#1:

The authors have addressed most of my concerns to my satisfaction. There are a few remaining concerns, which I hope can be addressed before publication.

Thank you for the nice comments. We considered all your criticism very seriously and have tried our best to address all your concerns. We hope you are satisfied with the 2nd revised version.

Q1: Naming Ga1P variants. There isn't an established nomenclature for CNV, but ZmGa1P is a gene with a name. The names should make it clear that these are Ga1P variants, not new pectin methylesterase genes. Giving the five ZmPmes-m genes a numerical designation is too similar to conventional PME gene names from other organisms. For example, ZmPme3 is a PME38 homolog and ZmGa1P is a PME63 homolog. Maybe something like Ga1P α , β (just a suggestion. As I said there is currently no convention for CNV).

[Response]: We greatly appreciate your thoughtful suggestion about the name of five PME genes. We agree that the five PME genes are homologs of *ZmGa1P*. Therefore, we now suggest to change the gene names to *ZmGa1P.1*, *ZmGa1P.2*, *ZmGa1P.3*, *ZmGa1P.4*, *ZmGa1P.5*, and uniformly called them as the *ZmGa1Ps-m* in the article. As for other PME genes located in the Component1 region, and since they are different from *ZmGa1P* in sequence and expression pattern, we adopted the nomenclature of *ZmPME4-(1-7)* to indicate that they are the PME gene located on maize chromosome4 to distinguish them from *ZmGa1P* and *ZmGa1Ps-m* (**Supplementary Fig. 5**). We hope you are satisfied with our modifications.

Supplementary Fig. 5 | Gene structure of thirteen *PME* genes of maize and expression levels in nine SK tissues. (a) Gene structure of thirteen *PME* genes. White boxes represent coding sequences. The *PME* domain are highlighted with red boxes, and UTR regions are highlighted with grey boxes. **(b)** Expression levels of thirteen *PME* genes in nine SK tissues. FPKM, Fragments Per Kilobase per Million mapped reads.

Q2: I would like to see discussion as to why Zhang 2018 detected one protein in SDG25a but Hp301 and SK seem to have 5 genes/copies. Is this a technical or biological difference? Does SDG25a have these 5 and the Zhang methods didn't catch them? Or is there a second allele for Ga1-S?

[Response]: Thank you for the nice comments. We revised the method in detecting proteins described by Zhang et al, 2018, which is based on LC-MS/MS analyses (Ref. 1). Fragmented peptides are obtained firstly, and then assembled according to the reference genome. The reference genome used in Zhang's study is a 135 Kb BAC sequence, containing only one complete *PME* gene, *ZmGa1P*. Therefore, we assumed that limited by the incomplete reference sequence, authors failed to assemble all the peptide fragments from multiple highly similar *PME* genes in the *Ga1* locus. Genome annotation of SK and HP301 were based on a high-quality *de novo* assemble strategy, in which contig N50 size is 15.78 Mb (Ref. 2) and 25.7 Mb (Ref. 3), respectively. Additionally, a comprehensive annotation strategy, combining *de novo* gene prediction, protein-based homology searches. RNA-Seq and Iso-Seq of multiple tissues was used to annotate genes, therefore, multiple *PME* genes can be found in an interval close to 1.5 Mb. Meanwhile, considering all materials identified as *Ga1-S* type in the AMP exhibit enriched reads count of the five *PME* genes (from AMP 20 × depth re-sequencing data), we assumed that there also exist multiple *PME* genes in SDG25a, Following reviewer's suggestion, we have added this description and write "Therefore, we assumed that there are also multiple copies of *ZmGa1P* in the SDG25a genome. However, limited by the length of the BAC clone, it is hard to assemble all the peptide fragment from multiple highly similar *ZmGa1P* genes in an interval close to 1.5Mb."

Q2: Supplementary Figure 16 (and 13): CML121 is known to be Ga1-M and CIMBL37 is known to be ga1. Are they mislabeled in your figure or do you have new data to suggest they are Ga1-S and Ga1-M, respectively?

[Response]: In order to accurately identify the type of AMP, we performed a fertility test in the field. For the CML121 and CIMBL37, we once again confirmed their field

phenotypes. We found both CML121 and CIMBL37 pollen lead to full seed set on CML304 (*Ga1-S*) silks. Zheng58 (*ga1*) pollen leads to full seed set on CIMBL37 silks, CML121 plants could not be pollinated by Zheng58 pollen (**Response Fig.1**). At the same time, based on the re-sequencing data of AMP, we verified that *ZmPRP3* is absent in CML121, while the *ZmGa1Ps-m* is present in CIMBL37. Therefore, we identified CML121 as *Ga1-S* type and CIMBL37 as *Ga1-M* type. As we know, AMP are widely used by scientists working on *Zea mays* all over the world, and what we discuss here can only represent AMP cited from Yang et al. 2010 (**Ref. 4**)

Response Fig.1 | Crossing fertility test. a, Crossing experiments showing ears of CML121 pollinated by Zheng58 (*ga1*) and ears of CML304 pollinated by CML121. **b**, Crossing experiments showing an ear of CIMBL37 pollinated by Zheng58 (*ga1*) and ears of CML304 pollinated by CIMBL37. Scale bar = 2 cm.

Q3: Line 240-241: need to add in a caveat that not ALL *Ga1-M* lines do have *ZmPRP3*.

[Response]: We greatly appreciate your suggestion. We found that there are 82 inbred lines that can be identified as *Ga1-M* type in the AMP, of which 9 inbred lines have *ZmPRP3* reads count less than 10. We tried to amplify *ZmPRP3* in 5 of 9 inbred

lines: CML360, CIMBL11, P138, CIMBL151, CML50, while we were not able to detect *ZmPRP3* in CML360, CIMBL11 and CML50. (**Response. Fig2**). Therefore, following reviewer's suggestion, we have added the description of the distribution of *ZmPRP3* in *Ga1-M*, that is, about 90% of *Ga1-M* type lines exhibit enriched reads count of *ZmPRP3*. We hope that you are satisfied with our modifications.

Response Fig.2 | Amplifying *ZmPRP3* in eleven *Ga1-M* and two *ga1* inbred lines.

Q4: Supp Figure 5: PME10 what is the relationship to ZmPME10-1 reported in Zhang 2018? Is this a different gene or is it mislabeled?

[Response]: Thank you for picking this. ZmPME10-1 is located on chromosome 10, but it is not the same gene as *ZmPME10*. According to the reviewer's comment **Q1**, we have revised the name of each gene in the PME gene cluster, and for clarity the name of *ZmPME10* has been revised to *ZmPME4-6*.

Q5: Lines 159-161: However, in the corresponding Component1 region of B73, there is only one gene (Zm00001d048936), encoding a PME and exhibiting synteny (Fig. 1c, d). This isn't wholly accurate. B73 has PME pseudogenes of both *Ga1P* and *ZmPme3* (Zhang 2018 and Moran Lauter 2017).

[Response]: We greatly appreciate your comment. We rechecked the annotated gene information of the B73 genome v4 version, and found that Zm00001d048936 was the only gene annotated as a PME in the Component1 region, Meanwhile, we refer to the description from Zhang et al., 2018 and corrected it and write "However, in the corresponding Component1 region of B73, there is only one annotated gene,

Zm00001d048936, (also known as the *ga1* type *ZmGa1P*), which is highly expressed in pollen and exhibits synteny to the PME gene cluster”.

Q6: Grammatical corrections:

- (1) Line 233: which encodes a small protein containing a barwin
- (2) Line 410: pollen with
- (3) Line 412: tube growth through silks
- (4) Lines 419-20: The presence of *Zmprp3* may explain why a few *Ga1-M* lines with intact *ZmPme3* genes and *ZmPme3*^{OE+} can still be fertilized by *ga1* pollen...
- (5) Line 505: types of maize were
- (6) Line 550: *Ga1-S* pollen grows through the *Ga1-S* silk
- (7) Line 557: it is possible

[Response]: Many thanks for picking the typos, we have carefully revised the whole text, including the seven parts pointed by the reviewer. We tried our best to correct the grammatical error. We hope you are satisfied with our modifications.

Response to Reviewer#2

The authors have responded and addressed the questions raised by the reviewers.

Thank you for the nice comments. We are happy that you are almost satisfied with our revision, we considered your criticism very seriously and have tried our best to address your concerns, we hope that the revised version can satisfy you.

Q1: I would say there is still something the authors should double check and it is the use of the t-test. I think that will very much depend on the type of data population they have, which they haven't described in their use of this method.

[Response]: We greatly appreciate your suggestions. We re-examined the datasets that we used to perform variance analysis. After performing Shapiro-Wilk normality tests, we found that most of the datasets conformed to the normal distribution. In order to explain the test method more rigorously, we used Wilcoxon ranked sum non-parametric test to recheck the conclusions of all variance analysis, and found that the conclusions were consistent with the previous ones. We summarized the results of Shapiro-Wilk normality test and Wilcoxon ranked sum test into the **Supplementary Table 8** to provide additional information, and added the description in the Method section. We hope you are satisfied with our modifications.

Response to Reviewer#3

I reviewed a modified version of the paper by Wang and collaborators on the genetic determinants of the unilateral cross incompatibility (UCI) in the genus *Zea*. The authors have deeply modified their manuscript and answered my comments. I would like to acknowledge their efforts in addressing the reviewers' comments. I have however two minor comments remaining

Thank you very much. We considered all your criticism very seriously and have tried our best to address your concerns, we hope that the revised version can satisfy you.

Q1: Regarding the Figure 1b, I still think that the numbers presented in the Table are not clearly explained in the legend. The authors should mention from which cross these progenies are coming from, what the counts represent.

[Response]: We appreciate this comment and apologize for the inconvenience of reading due to insufficient description in the legend. We have now added the description of counts and the source of progeny to avoid any misunderstandings.

Q2: As for Figure 4 which is a new figure that answers my first comments: I am asking them to modify the Figure 4c by discarding the use of arrows. The authors have shown that *parviglumis* encompasses two different versions of haplotypes, but they can't infer evolutionary paths. For example, *Ga1-M* (+++) could have evolved from (+++) or (---), and I see no reasons to think that the former is more likely. So, I would stay careful about the interpretations, and simply present *Teosintes* as encompassing two haplotypes, *Landraces* as two, and *Modern maize* as three without trying to infer an evolutive scenario. For example, it is not clear to me why the authors define the (+++) as ancestral in Figure 4d. To define an ancestral state one would need an outgroup species. Along this line, it would be good to discuss what are *par1* and *par2* (these clusters have likely been already described in the literature so it would help if the authors can relate them to what was discovered before, which of *par1* or *par2* is the cluster that corresponds to the putative domestication location?).

[Response]: We apologize that the evolutionary path has not been adequately described and explained in the previous version. We will explain the evolutionary path as follows.

(1) From the genome alignment results of SK (*Ga1-S*) and B73 (*ga1*), we concluded that there is a huge genome variation between *Ga1-S* and *ga1*, and from the fact that the three types of genes maintain fairly consistent haplotypes among *par1*, *lan1* and *ga1*, this mutation effectively inhibits the occurrence of recombination.

(2) Haplotype of the *Ga1* locus (about 2Mb region) that derived from *Ga1-M* (+++) exhibit the closet relationship to *Ga1-S* and *lan2* in PCA analysis, but distinguished from *ga1* and *lan1*. Meanwhile, *ZmGa1Ps-m* genes were absent in all *ga1* but present in *Ga1-M* and *Ga1-S* lines, *ZmPME3* haplotype of *Ga1-M* and *Ga1-S* is also very different from *par1*, *lan1* and *ga1*. Therefore, we assumed that *Ga1-M* and *ga1* have different origins, that is to say, *Ga1-M* cannot be derived from *par1* and *lan1*.

(3) *Ga1-M* retaining the high copy number of the *ZmGa1Ps-m* genes, and possess *ZmPRP3*, which is consistent with *par2* and *lan2*. A part of materials from *par2* and *lan2* exhibit a haplotype very similar to *Ga1-M* and *Ga1-S* in the *ZmPME3* gene region. Based on the above results, we assume that the origin of *Ga1-M* and *Ga1-S* is close and related to *par2* and *lan2*.

(4) Re-sequencing and quantitative RT-PCR results showed that *ZmPME3* in *Ga1-M* was not lost, but a 1 or 2 bp insertion occurred, resulting in loss of function, which is fundamentally different from the variation of *ZmPME3* in *ga1*. We did not even detect the complete gene sequence of *ZmPME3* on the B73 genome, nor the expression level of *ZmPME3* in the *ga1* population, although we labeled *ZmPME3* in both *Ga1-M* and *ga1* as “-”, it only means that *ZmPME3* lost function, does not represent their haplotype are the same. In order to show these conclusions more intuitively, we showed the sketch haplotype map of the *Ga1* locus across cultivated, landrace and wild maize and added it into **Supplementary Fig. 29** to show the gene haplotype change process and avoid any misunderstanding.

(5) *Zea mays* was domesticated from the wild grass *Zea mays ssp. parviglumis* approximately 9,000 years ago in the southwest of Mexico. For a more precise explanation, we now modified “ancestor haplotype” to “wild teosinte haplotype” here to refer to the haplotypes of *parviglumis* and *mexicana*.

(6) Following reviewer’s suggestion, we collected the geographic distribution of the two sub-groups of *parviglumis* (Ref.5), and found that the distribution of the two sub-groups is relatively concentrated, and there is no separation (**Response Fig. 3**). Therefore, we assume the divergence of the *Ga1* locus in *par1* and *par2* is more likely to be reflected in the genetic relationship or genome structure variation, which requires more data to support but exceed the scope of this manuscript.

We hope you are satisfied with our explanations.

Supplementary Fig. 29 | The sketch haplotype map of three types of genes of the *Ga1* locus on cultivated, landrace and wild maize genome. (a) *ZmPME3*, *ZmGa1Ps-m* and *ZmPRP3* on *par2*, *mexicana* (*mex*), *lan2*, *Ga1-M* and *Ga1-S* type genome. Five *ZmGa1Ps-m* genes, one silk-expressed gene *ZmPME3*, and *ZmPRP3* are highlighted by dark green, light green and orange boxes. Absent or incomplete genes are indicated by dotted boxes. (b) *ZmPME3*, *ZmGa1P^{ga1}* and *ZmPRP3* on *par1*, *lan1* and *ga1* type genome. *ZmGa1P^{ga1}*, *ZmPME3*, and *ZmPRP3* are highlighted by dark green, light green and orange boxes. Absent or incomplete genes are indicated by dotted boxes.

Response Fig.3 | Geographical distribution of the two sub-clusters of the *parviglumis* populations.

Q3: Finally, I found the Figure 4d quite confusing. I suggest that the authors use Figure 4c instead to report the frequency of each haplotype.

[Response]: We appreciate your suggestion. The evolution path described in Figure 4c indicates a hypothesis, but all proportions in Figure 4d are based on real and accurate data from the association mapping panel, including field tests and re-sequencing data. Therefore, we think adding the proportions of the gametophyte can provide more information and help readers to better understand the distribution of three different haplotypes in the *Ga1* locus and the rare haplotype. We have added more details to the legend to explain the frequency to avoid any misunderstandings.

Reference

1. Zhang, Z. et al. A PECTIN METHYLESTERASE gene at the maize *Ga1* locus confers male function in unilateral cross-incompatibility. *Nat. Commun.* **9**, 3678 (2018).
2. Yang, N. et al. Genome assembly of a tropical maize inbred line provides insights into structural variation and crop improvement. *Nat. Genet.* **51**, 1052-1059 (2019).
3. Hufford, M. B. et al. De novo assembly, annotation, and comparative analysis of 26 diverse maize genomes. *Science* **373**, 655-662 (2021)
4. Yang, X. H. et al. Characterization of a global germplasm collection and its potential utilization for analysis of complex quantitative traits in maize. *Molecular Breeding* **28**, 511-526 (2010).

5. Chen, L. et al. Portrait of a genus: the genetic diversity of *Zea*. *bioRxiv*. (2021).

Reviewers' Comments:

Reviewer #1:

Remarks to the Author:

I am satisfied with the revisions made. I think this manuscript is greatly improved, thank you. I would recommend the revised version for publication.

Reviewer #2:

Remarks to the Author:

The authors have addressed the concerns in a satisfactory way.

Reviewer #3:

Remarks to the Author:

I am fully satisfied with the reviewers responses and thank them again for their careful consideration of my comments.